# Implementation of a Deep Learning Algorithm Based on Vertical Ground Reaction Force Time–Frequency Features for the Detection and Severity Classification of Parkinson’s Disease

**DOI:** 10.3390/s21155207

**Published:** 2021-07-31

**Authors:** Febryan Setiawan, Che-Wei Lin

**Affiliations:** 1Department of Biomedical Engineering, College of Engineering, National Cheng Kung University, Tainan City 701, Taiwan; febryans2802.wtmh@gmail.com; 2Medical Device Innovation Center, National Cheng Kung University, Tainan City 701, Taiwan

**Keywords:** deep learning, gait analysis, Parkinson’s disease severity stages, time–frequency spectrogram, vertical ground reaction force (vGRF) signal

## Abstract

Conventional approaches to diagnosing Parkinson’s disease (PD) and rating its severity level are based on medical specialists’ clinical assessment of symptoms, which are subjective and can be inaccurate. These techniques are not very reliable, particularly in the early stages of the disease. A novel detection and severity classification algorithm using deep learning approaches was developed in this research to classify the PD severity level based on vertical ground reaction force (vGRF) signals. Different variations in force patterns generated by the irregularity in vGRF signals due to the gait abnormalities of PD patients can indicate their severity. The main purpose of this research is to aid physicians in detecting early stages of PD, planning efficient treatment, and monitoring disease progression. The detection algorithm comprises preprocessing, feature transformation, and classification processes. In preprocessing, the vGRF signal is divided into 10, 15, and 30 s successive time windows. In the feature transformation process, the time domain vGRF signal in windows with varying time lengths is modified into a time–frequency spectrogram using a continuous wavelet transform (CWT). Then, principal component analysis (PCA) is used for feature enhancement. Finally, different types of convolutional neural networks (CNNs) are employed as deep learning classifiers for classification. The algorithm performance was evaluated using *k*-fold cross-validation (*k*foldCV). The best average accuracy of the proposed detection algorithm in classifying the PD severity stage classification was 96.52% using ResNet-50 with vGRF data from the PhysioNet database. The proposed detection algorithm can effectively differentiate gait patterns based on time–frequency spectrograms of vGRF signals associated with different PD severity levels.

## 1. Introduction

Parkinson’s disease (PD) is a neurodegenerative disease that belongs to a group of motor system disorders caused by the loss of dopamine-producing brain cells. PD is the second most common neurodegenerative disease [1]; its prevalence is approximately 0.3% in the general population, approximately 1% in individuals older than 60, and approximately 3% in people aged 80 and over [1]. The incidence of PD is 8–18 per 100,000 people. The median age at onset is 60 years, and the mean duration of the progression of the disease from diagnosis to death is approximately 15 years [1]. There is a 1.5–2-fold greater prevalence and incidence of this disease in men [1]. PD treatments cost approximately USD 2500 each year, and therapeutic surgery costs up to USD 100,000 per patient [2]. The primary PD symptoms are tremors in the hands, arms, legs, jaw, and face; rigidity (inflexibility of the limbs and trunk); bradykinesia (slowness in movement); and postural instability (balance and coordination disturbance) [3,4,5]. As these symptoms become more severe, patients may experience difficulties walking, talking, or accomplishing simple tasks. Currently, there are no blood or laboratory tests that assist in diagnosing PD. Symptoms of the disease include certain characteristic walking difficulties, such as a shortened stride length, decreased gait speed, increased stride-to-stride variation, a shambling gait, and frozen gait.

Gait analysis is used to assess and treat individuals with conditions affecting their ability to walk, such as poor health, advanced age, size, weight, and speed. A standard assessment is needed to clinically identify and evaluate gait characteristics and other phenomena in PD patients, such as gait count, walking speed, and step length. Pistacchi et al. analyzed temporal parameters (see Figure 1) in patients with early PD using 3D gait analysis-related cadence (PD patients: 102.46 ± 13.17 steps/min and healthy subjects: 113.84 ± 4.30 steps/min), stride duration (PD patients: 1.19 ± 0.18 s right limb and 1.19 ± 0.19 s left limb; healthy subjects: 0.426 ± 0.16 s right limb and 0.429 ± 0.23 s left limb), stance duration (PD patients: 0.74 ± 0.14 s right limb and 0.74 ± 0.16 s left limb; healthy subjects: 1.34 ± 1.1 s right limb and 0.83 ± 0.6 s left limb), and velocity (PD patients: 0.082 ± 0.29 m/s; healthy subjects: 1.33 ± 0.06 m/s) [5]. Sofuwa et al. concluded that individuals with PD showed a significant reduction in step length and walking speed compared with the non-PD control group [6]. These observations suggest that foot force is affected by PD. Lescano et al. aimed to analyze gait parameters, stance, swing phase duration, and the magnitude of the vertical component of the ground reaction force for the purpose of assessing whether there are statistically significant differences between PD patients in stages 2 and 2.5 (modified Hoehn and Yahr (HY) scale, see description in Table 1) [7]. Gait information has been developed for movement analysis in healthy control (CO, term defined by PhysioNet [8]) subjects and other subjects with different types of diseases. This approach is useful for understanding movement disorders arising from PD, and may be valuable in developing non-invasive automatic detection and severity classification approaches for PD.

Classification is the process of identifying the class of a new observation using a set of categories based on a training process involving observations for which the classes are known. In PD classification, various machine learning algorithms have been implemented as classifiers and combined with sophisticated feature extraction methods for dimensionality reduction. Recently, deep learning approaches, instead of conventional machine learning algorithms, have been applied to improve PD classification performance. For example, Jane et al. presented a Q-backpropagated time delay neural network (Q-BTDNN) in a clinical decision-making system (CDMS) to diagnose patients with PD (PD vs. CO) [11]. The Q-BTDNN was trained using a Q-learning induced backpropagation (Q-BP) training algorithm by generating a reinforced error signal, and the weights of the network were corrected through the backpropagation of the generated error signal. Correa et al. implemented a method to model PD patients’ difficulties in starting and ending movements by examining information from speech, handwriting, and gait [12]. These researchers trained a convolutional neural network (CNN) to classify PD patients and CO subjects. The PD population in the database was divided into three groups based upon the stage of PD: low, intermediate, or severe. Lee and Lim classified idiopathic PD patients and COs based on their gait force characteristics using a continuous wavelet transform (CWT) to generate approximate coefficients and detail coefficients [13]. Forty features were extracted from those coefficients using statistical approaches, including frequency distributions and their variabilities. The features of idiopathic PD patients and COs were classified using a neural network with weighted fuzzy membership functions (NEWFM). Zhao et al. developed a two-channel model that combined Long Short-Term Memory (LSTM) and CNNs to learn spatio-temporal patterns in gait data recorded by foot sensors [14]. The model was trained and tested on three public vGRF datasets. The model could perform multi-category classification on features such as the severity level of PD, while previous machine learning-based approaches could only perform binary classification.

As previously mentioned, only a few studies have used the deep learning approach for the detection and severity classification of PD, and some of them have used statistical features combined with machine learning methods. The drawbacks of using machine learning are the dependence of its performance on data size and the understanding of features [15,16]. Machine learning only performs well on small to medium datasets and needs a better understanding of features to represent the data. The objective of this work was to develop a deep learning classifier to help physicians screen and classify the severity of PD in patients, using vGRF spectrograms. The effectiveness of time–frequency spectrogram (feature transformation) of vGRF signals from left (LF), right (RF), and compound foot (CF = LF + RF) movements in classifying features of PD severity was investigated. Specifically, the aim was to determine whether a significant difference in vGRF is related to the specifics of disease severity, as passive (weight acceptance) and active (push off) peaks of vGRF are important gait parameters [17] and exhibit significant relevance in the detection of gait abnormalities, especially in the PD gait assessment [13,14,18,19,20]. Different deep learning algorithms (including AlexNet, ResNet-50, ResNet-101, and GoogLeNet) were also utilized with the proposed method to compare the effectiveness among classifiers.

## 2. Materials and Methods

The proposed PD severity classification algorithm attempts to extract pattern features and visualizations from vGRF signals in PD patients with severity stages of 0, 2, 2.5, and 3 on the HY rating scale by transforming one-dimensional time domain signals into two-dimensional patterns (images) using the feature transformation method from a CWT. The proposed PD severity classification algorithm consists of four main steps, as shown in Figure 2: (1) signal preprocessing of PD patients’ vGRF signals, (2) feature extraction from a spectrogram of the vGRF signal generated using CWT and PCA, (3) construction and training of a CNN classifier, and (4) cross-validation to evaluate the performance of the classification algorithm.

### 2.1. Gait in Parkinson’s Disease Database

The vGRF database used in this research, the Gait in Parkinson’s Disease Database (gaitpdb), is available online from PhysioNet [8]. The database comprises three datasets, which were contributed by Yogev et al. (Ga) [21], Hausdorff et al. (Ju) [22], and Frenkel-Toledo et al. (Si) [23,24].

The database contains information recorded from 93 idiopathic PD patients (average age: 66.3 years; 63% men and 37% women) and 73 CO subjects (average age: 66.3 years; 55% men and 45% women). Every subject was instructed to walk at their usual pace for about two minutes while wearing a pair of shoes with eight force sensors located under each insole. The raw vGRF signal data in this database were obtained using force-sensitive sensors (Ultraflex Computer Dyno Graphy, Intronic Inc., NL-7650 AB Tubbergen, The Netherlands) with the output proportional to the force under the foot in Newtons, collected at 100 samples per second (frequency of readings during movement was 100 Hz). The recordings also included two signals that reflect the sum of the eight sensor outputs from the left and right foot.

The database also contains information about each participant, including gender, age, height, weight, walking velocity, and severity level of PD. The PD severity level was assigned according to two rating scales, HY [10] and the Unified Parkinson’s Disease Rating Scale (UPDRS) [25]. The HY rating scale, widely used to represent the way in which symptoms of PD progress, defines five stages of PD, with two additional intermediate stages, 1.5 and 2.5 (Table 1) [10]. The number of participants diagnosed using the HY rating scale is shown in Table 2.

### 2.2. Signal Preprocessing

A two-minute foot force signal was acquired during data collection from subjects. The LF, RF, and CF vGRF signals of the CO and PD subjects were used as inputs to the proposed algorithm. It was difficult to interpret the foot force data directly, despite using a CWT to transform the features, due to the length of the foot force signal. To observe the foot force signal more accurately, a window function was employed. A window function is a mathematical construct that is zero-valued outside of selected intervals. In this research, 10, 15, and 30 s window sizes were used. The aim of the time-windowing process was to obtain shorter signal data. In the clinical application, this data collection is more convenient for the PD patient and, furthermore, reduces the fall risk. The possibility of patient injury rises if the data collection time is longer. Normalization and zero-mean processing were also used, to reduce the redundancy and dependency of data.

In 1987, Nilsson and Thorstensson observed the adaptability in the frequency and amplitude of leg movements during human locomotion at different speeds [26]. They reported that the overall range of stride frequency for normal leg movements is 0.83–1.95 Hz. The stride cycle period is defined as the time from the heel contact of one foot with the ground to the next heel contact of the same foot with the ground. The stride cycle period can be derived from the vGRF signal, and the stride frequency is the inverted value of the stride cycle duration. In conclusion, we selected two frequency ranges, 0.83–1.95 Hz and 1.95–50 Hz, for detailed observations of vGRF spectrograms among CO and PD subjects.

### 2.3. The Continuous Wavelet Transform

The continuous wavelet transform (CWT) is a signal processing tool used to observe the time–frequency spectrum characteristics of non-stationary signals [27]. As in the case of the Gabor transform [28], a CWT can be used to filter a signal using a dilated version of the mother wavelet, but the frequency translation is affected by dilation (scaling) and contraction. A CWT is a time–frequency transformation method, because it changes the signal time domain to the time–frequency domain. The output of a CWT is a time–frequency spectrogram (time–scale representation), which provides valuable information about the relationship between time and frequency.

A CWT consists of a time series function $x\left(t\right)\in L^{2}(R)$, with a scaling or dilation factor $s\in R^{+}\;(s>0)$ that controls the width of the wavelet, and a translation parameter, τ, controlling the location of the wavelet as expressed in the following equation:$$X_{w}\left(s,\tau \right)=\frac{1}{\sqrt{s}}\overset{\infty }{\underset{-\infty }{\int }}x\left(t\right)\psi^{*}\left(\frac{t-\tau }{s}\right)dt$$
where $\psi \left(t\right)$ is a mother wavelet, also called a window function. The mother wavelet function used in this research was a Morlet or Gabor wavelet. This wavelet function consists of a Gaussian-windowed complex sinusoid (a complex exponential multiplied by a Gaussian window) as follows:$$\psi_{\omega_{0}}\left(t\right)=\left(e^{-ift}-e^{-\frac{1}{2}f^{2}}\right)e^{-\frac{1}{2}t^{2}}$$

The parameter t refers to the time, and f represents the reference frequency.

The aim of the time–frequency transformation is to represent the vGRF signal (Figure 3a) as a time–frequency spectrogram image, as shown in Figure 3b,c, Figure 4 and Figure 5. The images clearly show different patterns of vGRF between CO and PD subjects that cannot be found in the time and frequency domains of the signal. Using the time–frequency spectrogram, variations in the foot pressure signal caused by temporal characteristics can also be analyzed. Temporal characteristics, also known as spatial characteristics or linear gait variabilities, consist of the measurements of step length, the stance width, the length of the step rhythm, and the step velocity.

### 2.4. Principal Component Analysis

The main goal of principal component analysis (PCA) is to perform dimension reduction for a dataset containing a large number of interrelated variables while, to the greatest extent possible, retaining the variations present in the dataset [29]. This reduction is achieved by transforming the dataset into a new set of variables, the principal components (PCs), which are ordered, de-correlated variables.

The PCA method is defined mathematically using the following steps. Consider a matrix $X={\left[P_{1};P_{2};P_{3};\ldots ;P_{2}\right]}^{T}$ constructed using the spectrogram images of PDs and COs, where P is a row vector consisting of the pixels of a spectrogram image of PDs or COs, and i is the number of spectrogram images of PDs and COs. The PC is built using the equation $X^{T}X$, a covariance matrix of the matrix X, to determine its eigenvalues and eigenvectors. The W matrix, an m×m matrix of weights whose columns are the eigenvectors of $X^{T}X$, is obtained. Finally, the matrix for extracted feature F can be described as the full PCs’ decomposition of X, as shown in the following equation: F=XW.

The purpose of using PCA for feature enhancement was to extract fewer patterns while identifying the most important texture and pattern features. This processing was conducted in order to improve the performance of machine learning and artificial intelligence algorithms used for classifying the data points. The full PCs of each spectrogram image sample were selected to preserve the important texture and pattern features for visualization.

### 2.5. Convolutional Neural Network

A convolutional neural network (CNN) is composed of one or more convolutional layers, often with subsampling and pooling layers, followed by one or more fully-connected layers, as in a basic multi-layer neural network [30]. CNN was utilized to distinguish the time–frequency spectrogram representation of vGRF between PD severity stages. The convolutional layer plays the most important role in CNN performance. This layer is composed of a set of kernels (learnable filters) as parameters, which contain a small receptive field but are expanded through the full depth of the input. When the data pass through this layer, each kernel is convolved across the spatial dimensionality of the input (width and height of the input volume), resulting in the calculation of the dot product and production of a 2D activation map. The filters in the convolutional layers are edge detectors and color filters. An activation layer utilizes a non-saturating activation function $f\left(x\right)=\max\left(0,x\right)$, such as a sigmoid function, in which $\sigma \left(x\right)={\left(1+e^{-x}\right)}^{-1}$, to generate the output from the input produced by the previous layer. Another important concept in CNNs is pooling, also known as non-linear down-sampling. The aim of the pooling layer is to reduce the dimensionality and minimize the number of parameters and the complexity of model computation. This layer, known as the max-pooling layer, takes the input of each activation map and scales the input dimension using the MAX function. Finally, the fully connected layers attempt to generate scores from the previous activations to use for classification, as in traditional artificial neural networks (ANNs). Neurons in this layer have connections to all of the outputs of the previous layer. The performance of AlexNet, ResNet-50, ResNet-101, and GoogLeNet was examined in this study.

#### 2.5.1. AlexNet CNN

The AlexNet architecture [31] comprises 25 layers, including an input layer, 5 convolution 2D layers, 7 rectified linear unit (ReLU) layers, 2 cross-channel normalization layers, 3 max-pooling 2D layers, 3 fully connected layers, 2 dropout layers for regularization, a softmax layer using a normalized exponential function, and an output layer. The input to the AlexNet CNN in the proposed method is a time–frequency spectrogram of the vGRF signals produced by the CWT. There are two methods for fine-tuning a pretrained AlexNet CNN: transfer learning and feature extraction. We chose the feature extraction method because it is easy to apply to pretrained networks without expending a lot of time, as it is faster than the transfer learning method and requires less training. This method applies two previous fully connected layers and uses a support vector machine (SVM) for classification.

#### 2.5.2. ResNet-50 and ResNet-101 CNN

The main idea behind a residual network (ResNet) [32] is the presentation of a so-called “identity shortcut connection” that skips one or more layers. A shortcut (or skip) connection is used to solve the problem of vanishing or exploding gradients by using blocks that re-route the input and add to the concept learned in the previous layer. During learning, a layer learns the concepts of the previous layer and merges with inputs from that previous layer. ResNet-X refers to a residual deep neural network with X number of layers; for example, ResNet-50 indicates a ResNet developed using 50 layers. The architectures of ResNet-50 and ResNet-101 are described in Table 3.

#### 2.5.3. GoogLeNet CNN

GoogLeNet [33] is a pretrained CNN that has 22 layers with 9 inception layers. An inception layer determines the optimal local sparse structure in a convolutional vision network, which can be approximated and covered by readily available dense components. In general, the inception layer is a network consisting of parallel convolutions of different sizes and types (1×1, 3×3, and 5×5) for the same input, which stacks all of the outputs. The exact structure of GoogLeNet is as follows:An average pooling layer with a 5×5 filter size and a stride of 3.A 1×1 convolution with 128 filters for dimension reduction and rectified linear activation.A fully connected layer with 1024 units and rectified linear activation.A dropout layer with 70% rate of dropped outputs.A linear layer with softmax loss as the classifier.

Although AlexNet, ResNet-50, ResNet-101, and GoogLeNet achieved significant performance in the PD severity detection (overall accuracy ~97%), their architecture characteristics exhibited different influences on performance based on the benefits and drawbacks of the networks. The advantages and disadvantages of AlexNet, ResNet-50, ResNet-101, and GoogLeNet applied in the proposed method are summarized in Table 4.

### 2.6. Cross-Validation

Cross-validation is a statistical method used to assess and compare learning algorithms by dividing data into two groups: a training set used to train a model and a testing set used to test the model [36]. The training and testing sets are varied in consecutive rounds so that each data point is tested using a classifier in whose training it did not participate. There are two main purposes of using cross-validation. Cross-validation is used to quantify the generalizability of an algorithm, by testing the classifier on unseen data. The second purpose is to evaluate the performance of different algorithms and identify the best algorithm with which to classify the available data or, alternatively, to compare the performance of two or more variants of a parameterized model. In order to compare the results with the existing literature, k-fold cross-validation was utilized. Consequently, k iterations of training and testing were carried out in such a way that within each iteration, a different fold of the dataset was used for testing, while the remaining (k-1) folds were used for training. In this research, 10-fold cross-validation was applied.

## 3. Results

The experiments were carried out using MATLAB R2018a software on an NVIDIA GeForce GTX 1060 6 GB computer with 24 GB RAM. The computation time is affected by the number of input time–frequency spectrogram images (related to the time-windowing process, where a smaller time window will result in more images and longer computation time) and the number of neurons in the CNN. We employed multi-class classification for the COs and PD Stages 2, 2.5, and 3. This approach is representative of real-life applications, because doctors and neurologists do not have preliminary information about whether a patient is healthy or suffers from PD and, if the latter, what the severity is.

The sensitivity, specificity, accuracy, and AUC value of the proposed method were included as parameters for evaluation. The detailed definition of each evaluation parameter is provided in [37]. When selecting between diagnostic tests, Youden’s index is often applied to evaluate the effectiveness of the test [38]. Youden’s index is a function of sensitivity and specificity, and its value ranges between 0 and 1. A value close to 1 indicates that the diagnostic test’s effectiveness is relatively high and the test is close to perfect, and a value close to 0 indicates poor effectiveness, where the test is useless. Youden’s index (J) is the sum of the two fractions and indicates whether the measurements correctly diagnosed the diseased group (sensitivity) and healthy controls (specificity) over all cut-points c, −∞<c<∞:$$J=max_{c}\left\{\mathrm{sensitivity}\left(c\right)+\mathrm{specificity}\left(c\right)-1\right\}$$

### 3.1. PD Severity Classification of Separated Ga, Ju, and Si Datasets

The gaitpdb database [8] contains three different vGRF datasets based on different studies: the Ga dataset describes dual tasking in PD patients, the Ju dataset indicates rhythmic auditory stimulation (RAS) in PD patients, and the Si dataset represents treadmill walking in PD patients. There are 29 PD patients and 18 CO subjects in the Ga dataset, 29 PD patients and 26 CO subjects in the Ju dataset, and 35 PD patients and 29 CO subjects in the Si dataset. The input signal for the proposed algorithm was dependent on the window size during the time-windowing process. For the 10 s window, the input signal numbers for CO, PD Stage 2, PD Stage 2.5, and PD Stage 3 in the Ga, Ju, and Si datasets were 447, 492, 240, and 168; 199, 352, 460, and 109; and 348, 336, and 84, respectively. In the 15 s time window, the input signal numbers for CO, PD Stage 2, PD Stage 2.5, and PD Stage 3 in the Ga, Ju, and Si datasets were 297, 328, 160, and 112; 129, 229, 300, and 72; and 232, 224, and 56, respectively. In the 30 s time window, the input signal numbers for CO, PD Stage 2, PD Stage 2.5, and PD Stage 3 in the Ga, Ju, and Si datasets were 147, 164, 80, and 56; 58, 103, 135, and 35; and 116, 112, and 28, respectively.

The proposed method covered two kinds of classifications, multi-class (CO vs. PD Stage 2 vs. PD Stage 2.5 vs. PD Stage 3) classification and two-class (CO vs. PD) classification. In the two-class classification, PD Stage 2, 2.5, and 3 datasets were combined into one PD dataset. The best classification performance was obtained using AlexNet CNN for multi-class classification and ResNet CNN for two-class classification. The best classification result of the Ga dataset has 98.15% sensitivity, 98.16% specificity, 98.16% accuracy, and an AUC value of 0.9816 on average for multi-class classification and 99.77% sensitivity, 98.80% specificity, 99.11% accuracy, and an AUC value of 0.9995 for two-class classification. The best classification result of the Ju dataset has 98.06% sensitivity, 98.38% specificity, 98.24% accuracy, and an AUC value of 0.9822 on average for multi-class classification and 98.94% sensitivity, 99.04% specificity, 99.01% accuracy, and an AUC value of 0.9993 for two-class classification. The best classification result for the Si dataset has 97.73% sensitivity, 98.76% specificity, 98.27% accuracy, and an AUC value of 0.9825 on average for multi-class classification and 98.85% sensitivity, 98.41% specificity, 98.56% accuracy, and an AUC value of 0.9964 for two-class classification. Based on these classification results, the performance of the proposed method was not influenced by different datasets in the database, even though the data collection processes varied among these datasets.

### 3.2. PD Severity Classification of All Datasets (Merged)

For this classification, the three vGRF datasets in gaitpdb were merged and used as inputs to the proposed PD severity classification algorithm. For the 10 s, 15 s, and 30 s time windows, the input signal numbers for CO, PD Stage 2, PD Stage 2.5, and PD Stage 3 were 994, 1180, 784, and 277; 658, 781, 516, and 184; and 321, 379, 243, and 91, respectively. The best result for this classification type was obtained using the ResNet CNN, with 92.08% sensitivity, 95.60% specificity, 94.58% accuracy, and an AUC value of 0.9384 on average for multi-class classification and 94.46% sensitivity, 97.69% specificity, 96.63% accuracy, and an AUC value of 0.9949 for two-class classification. The complete classification results are shown in Table 5, Table 6, Table 7, Table 8, Table 9, Table 10, Table 11, Table 12, Table 13, Table 14, Table 15 and Table 16 for multi-class and Table 17, Table 18 and Table 19 for two-class, and Table 20, Table 21, Table 22 and Table 23 summarizes the classification results.

## 4. Discussion

In this section, we discuss the gait analysis for each severity stage of PD based on the time and frequency analyses of the time–frequency spectrograms. Some key features of a signal are difficult to observe with the naked eye, but time–frequency spectrogram analysis can help to decipher important information regarding time and frequency characteristics. A CWT was used in this study to transform the signal from the time domain into the time–frequency domain. The gait phenomena could be identified using pattern visualization and recognition based on time–frequency spectrograms for CO subjects and PD patients with severity stages of 2, 2.5, and 3.

This observation was only performed for the CF vGRF signal. Since this type of input signal is the additive force between the left and right foot force signals, it describes the correlations between the features of the left and right feet instead of a single feature of the left or right foot. In order to further investigate the gait phenomena, a 10 s time window spectrogram was selected because the image feature was derived from a shorter input signal, and more detail can be perceived from the texture and pattern visualization of gait phenomena. For a 15 and 30 s time window spectrogram, the texture and pattern information is more compressed, and thus, the gait phenomena are blurred and not easily observed (see Figure 4 and Figure 5). The 0.1–5 Hz and 5–50 Hz frequency ranges were only applied to the detailed observations of the CWT time–frequency spectrogram and were not used for the classification.

### 4.1. Healthy Controls

Normal gait phenomena were interpreted by observing the time–frequency spectrogram of CO subjects, as shown in the first column of Figure 3. In the 0.1–5 Hz frequency range (Figure 3, first column, second row), the strongest walking force magnitude, represented in yellow, of the normal gait occurs at 1.6–2.1 Hz and is stable from the initial time to the end of the experiment. The foot force distributions and walking velocities for normal subjects were therefore the same when they were walking. At 2.5–3 Hz and approximately 4.5–5 Hz, small areas signifying the lowest force magnitude, shown in dark blue, alternate with a significant force magnitude, indicated by light blue, forming a regular pattern. This phenomenon appears in the spectrogram and is caused by the CF force signal at the lowest magnitudes. The three lowest magnitudes can be observed in one cycle of the CF force time domain signal (top left of Figure 3); the lowest magnitudes are almost equal in every cycle of the signal. The lowest magnitudes that occur at the beginning and end of the half gait cycle (that is, only the left or right foot gait cycle), close to the 0 force unit, show that the toe-off and initial contact and the lowest magnitude that occur between the half gait cycle are demonstrated only when one foot is in contact with the ground.

In the 5–50 Hz frequency range (Figure 3, first column, third row), a steady, strong force level, represented in yellow, occurs at approximately 5 Hz, with the same magnitude as that which occurs during walking, from the beginning to the end of the recording, and a significant force magnitude, shown in light blue, occurs up to 50 Hz in all records. Both time–frequency spectrograms indicate that the time and frequency components in the spectrogram have a regular pattern. This interpretation became a benchmark for investigating PD gait phenomena. These data were compared to analyze the gait characteristics of PD patients based on spectrogram analyses.

### 4.2. Parkinson’s Disease Stage 2

The time–frequency spectrograms for PD patients were similar to those of the CO spectrograms. For PD Stage 2 patients, as presented in the second column of Figure 3, the strongest force is at 1.6–2.1 Hz in the 0.1–5 Hz (Figure 3, second column, second row) frequency range, and there is a significant, strong magnitude, shown in light yellow, at 1 Hz, which is weaker than the force magnitude at 1.6–2.1 Hz. The significant force magnitude at 2.5–3 Hz and approximately 4.5–5 Hz becomes more yellow instead of light blue as in the CO spectrogram. It is also apparent that the pattern of the lowest force magnitude at 2.5–5 Hz is regular at some times and irregular at other times. This observation indicates that the magnitudes of the global and local minima are not the same in every gait cycle (Figure 3, second column, first row). In the time domain, the CF vGRF signal has fluctuating force magnitudes that cause an irregularity in the signal.

In the 5–50 Hz frequency range (Figure 3, second column, third row), the strongest force magnitude, shown in yellow, is about 5 Hz, and significant force, represented by light blue, occurs up to 50 Hz every time. However, the force magnitude is not distributed equally over the entire walking period.

### 4.3. Parkinson’s Disease Stage 2.5

As shown in the third column of Figure 3, the spectrogram for PD Stage 2.5 patients is not very different from the PD Stage 2 spectrogram in either frequency range. The only difference is that, in the 0.1–5 Hz frequency range, a significant, strong magnitude at 1 Hz becomes stronger, and yellow areas of force magnitude appear in the image. PD patients in the early stages—2 and 2.5—of the disease can have a walking velocity similar to that of COs, but their force distribution is typically not equally distributed, due to the presence of tremors.

### 4.4. Parkinson’s Disease Stage 3

Of the patients studied in this research, those with PD Stage 3 had the most severe level of disease. The spectrograms of this group exhibit the most irregular patterns of all severity levels. In the fourth column, first row of Figure 3, the CF vGRF signal has the most fluctuation and irregular force magnitudes because of the jerky movements and tremors of the patients.

In the 0.1–5 Hz frequency range (third column, second row of Figure 3), the strongest walking force magnitude, shown in yellow, occurs at a lower frequency than in stages 2 and 2.5 at 1–1.5 Hz. A significant strong force magnitude, depicted in light yellow, also appears at approximately 0.75 Hz, although the force level is not the same in every gait cycle. At 2–3 Hz and 3.5–4 Hz, significant force magnitude regions occur, as shown by colors that are more yellow.

In the 5–50 Hz frequency range (third column, third row of Figure 3), the strongest force magnitude only appears in certain gait cycles and is not equally distributed. A significant force magnitude, shown in light blue, only occurs up to 20 Hz, and forms an irregular pattern in every gait cycle.

### 4.5. Comparison of Results with the Existing Literature

A comparison between the proposed methodology and a study by Zhao et al. [14] is presented in Table 24. The authors carried out multi-class classification of vGRF signals for CO vs. PD Stage 2 vs. PD Stage 2.5 vs. PD Stage 3 using the same information found in the same database used for the proposed method, gaitpdb. These authors separated the classification types based on the three datasets—Ga, Ju, and Si—and used 10-fold cross-validation as the evaluation method. The two-class classification results were also compared with those of studies conducted by Maachi et al. [39], Wu et al. [40], Ertugrul et al. [41], Zeng et al. [42], Daliri [43], and Khoury et al. [44,45]. These comparison results are shown in Table 25 and Table 26. In Khoury et al.’s study, the classification types were divided based on the three datasets—Ga, Ju, and Si.

In summary, the proposed method produced almost the same classification results as those published in the existing literature, but the proposed algorithm generated better visualizations via time–frequency spectrograms associated with the progression of PD severity. The irregularity in patterns in the spectrograms is proportional to the severity level of PD. The more severe the disease, therefore, the more irregular the spectrogram’s pattern. This phenomenon could be helpful for medical specialists or neurologists in monitoring PD progression, allowing them to provide more effective and accurate medications and therapies to patients.

## 5. Conclusions

In this study, a deep learning algorithm was implemented based on vGRF time–frequency features for the detection and severity classification of Parkinson’s disease. Pattern visualization and recognition of the time–frequency spectrogram made it possible to successfully differentiate PD severity stages and COs. A CWT was used to generate spectrograms to visualize gait foot force signals by transforming signals from the time domain into the time–frequency domain. Three time-window sizes (10, 15, and 30 s), two frequency ranges (0.83–1.95 and 1.95–50 Hz), and three types of gait foot force signals (LF, RF, and CF force signals) were selected as inputs to obtain good feature visualization. After the original signal was transformed, PCA was applied for feature enhancement, to increase between-class separability and to reduce within-class separability. Finally, CNNs were used to perform classification. To evaluate the CNNs’ classification process, 10-fold cross-validation was performed, and the accuracy, sensitivity, specificity, and the AUC value were evaluated. The proposed method was able to achieve the highest performance for more than 97.42% of the parameters being evaluated and achieved superior performance in comparison with the detection and PD severity classification performance of state-of-the-art methods found in the literature.

Although the evidence indicates that the proposed method achieved good performance, there are several major drawbacks that could be improved. First, an existing database was used with the proposed method, and clinical data with a greater number of severity levels should be used to verify the performance and to resolve the limitation of the relatively small number of PD patients at certain severity levels in the current database. Clinical data collection will be carried out using a smart insole with an embedded 0.5” force sensing resistor of our own design. The precision and accuracy of force sensing resistor readings are also considered in order to obtain the correct representation of the vGRF signal. PD patients will be asked to perform some simple daily activities, such as turning around and sitting, instead of only walking down a long pathway. Second, long-term data collection to monitor PD progression is important for treatment decisions, since the gait patterns of PD patients appear to change with the long-term progression of the disease. Third, to further investigate the clinical meaning of the results, PD gait phenomena based on time–frequency spectrograms should be discussed with physicians. Fourth, other input data, such as kinetic data, temporal data, step length, and cadence, and other classifiers should be used to confirm and compare the effectiveness of pattern visualization and recognition based on the use of time–frequency spectrograms in PD detection.

## Figures and Tables

**Figure 1 sensors-21-05207-f001:**
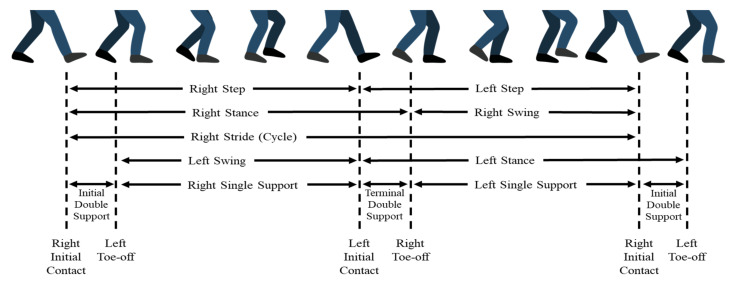
Description of beginning and ending of different gait phases in a normal gait cycle (Arafsha et al. [9]).

**Figure 2 sensors-21-05207-f002:**
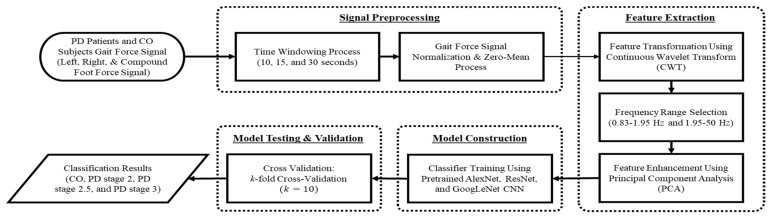
Flowchart of the proposed PD detection and severity classification algorithm using the continuous wavelet transform as the feature transformation.

**Figure 3 sensors-21-05207-f003:**
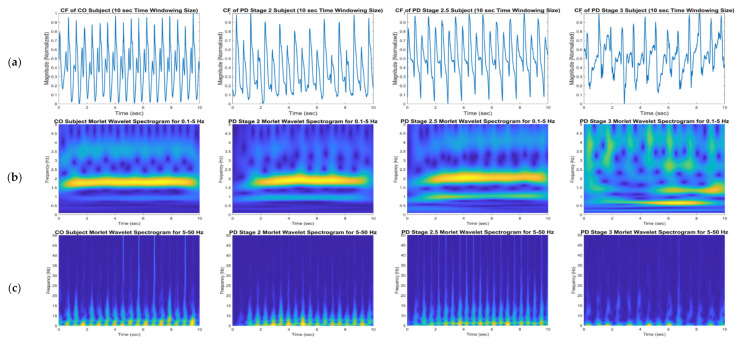
Time-frequency spectrograms using CWT of the first 10 s CF vGRF signals of CO, PD Stage 2, PD Stage 2.5, and PD Stage 3 subjects in a 10 s time window size from Ga [21] dataset: (**a**) original vGRF signal, (**b**) 0.1–5 Hz time-frequency spectrogram, and (**c**) 5–50 Hz time-frequency spectrogram.

**Figure 4 sensors-21-05207-f004:**
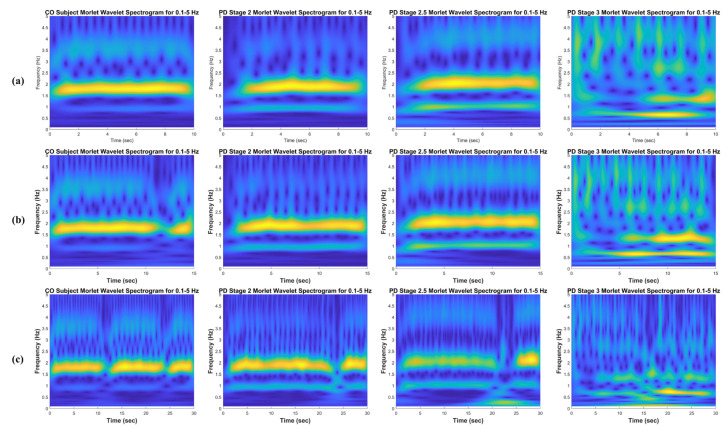
Time-frequency spectrograms using CWT of the CF vGRF signals of CO, PD Stage 2, PD Stage 2.5, and PD Stage 3 subjects in the 0.1–5 Hz frequency range for (**a**) a 10 s time window size, (**b**) a 15 s time window size, and (**c**) a 30 s time window size from Ga [21] dataset.

**Figure 5 sensors-21-05207-f005:**
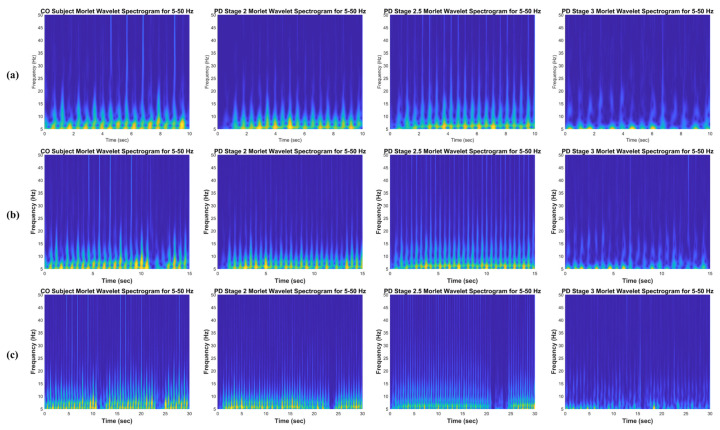
Time-frequency spectrograms using CWT of the CF vGRF signals of CO, PD Stage 2, PD Stage 2.5, and PD Stage 3 subjects in the 5–50 Hz frequency range for (**a**) a 10 s time window size, (**b**) a 15 s time window size, and (**c**) a 30 s time window size from Ga [21] dataset.

**Table 1 sensors-21-05207-t001:** Hoehn and Yahr (HY) scale [10] for PD severity stage.

Stages	Description
0	No signs of disease
1	Symptoms are very mild; unilateral involvement only
1.5	Unilateral and axial involvement
2	Bilateral involvement without impairment of balance
2.5	Mild bilateral disease with recovery on pull test
3	Mild to moderate bilateral disease; some postural instability; physical independence
4	Severe disability; still able to walk or stand unassisted
5	Wheelchair bound or bedridden unless aided

**Table 2 sensors-21-05207-t002:** Number of subjects in three sub-datasets of Parkinson’s Disease database (gaitpdb) [8] based on the HY rating scale of severity.

Author	Stage 0	Stage 2	Stage 2.5	Stage 3
Ga [21]	18	15	8	6
Ju [22]	26	12	13	4
Si [23]	29	29	6	0

**Table 3 sensors-21-05207-t003:** Architectures for ResNet-50 and ResNet-101.

Layer Name	Output Size	50 Layer	101 Layer
conv1	112×112	7×7, 64, stride 2	
conv2_x	56×56	3×3 max pool, stride 2	
		$\left[\begin{array}{c} 1\times 1,\;64\\ 3\times 3,\;64\\ 1\times 1,\;256 \end{array}\right]\times 3$	$\left[\begin{array}{c} 1\times 1,\;64\\ 3\times 3,\;64\\ 1\times 1,\;256 \end{array}\right]\times 3$
conv3_x	28×28	$\left[\begin{array}{c} 1\times 1,\;128\\ 3\times 3,\;128\\ 1\times 1,\;512 \end{array}\right]\times 4$	$\left[\begin{array}{c} 1\times 1,\;128\\ 3\times 3,\;128\\ 1\times 1,\;512 \end{array}\right]\times 4$
conv4_x	14×14	$\left[\begin{array}{c} 1\times 1,\;256\\ 3\times 3,\;256\\ 1\times 1,\;1024 \end{array}\right]\times 6$	$\left[\begin{array}{c} 1\times 1,\;256\\ 3\times 3,\;256\\ 1\times 1,\;1024 \end{array}\right]\times 23$
conv5_x	7×7	$\left[\begin{array}{c} 1\times 1,\;512\\ 3\times 3,\;512\\ 1\times 1,\;2048 \end{array}\right]\times 3$	$\left[\begin{array}{c} 1\times 1,\;512\\ 3\times 3,\;512\\ 1\times 1,\;2048 \end{array}\right]\times 3$
	1×1	average pool, 1000-d fc, softmax	
FLOPs		$3.8\times 10^{9}$	$7.6\times 10^{9}$

**Table 4 sensors-21-05207-t004:** The advantages and disadvantages of the AlexNet, ResNet, and GoogLeNet CNN architecture [34,35].

Architecture	Advantage	Disadvantage
AlexNet layer depth: 8 parameters: 60 million	There is low feature loss, as the ReLU activation function does not limit the output. Uses data enhancement, dropout, and normalization layers to prevent the network from overfitting and improve the model generalization.	This model has much less depth; hence, it struggles to learn features from image sets. Takes more time to achieve higher accuracy (highest accuracy achieved: 99.11%).
ResNet-50 layer depth: 50 parameters: 25.6 million ResNet-101 layer depth: 101 parameters: 44.5 million	Decreased the error rate for deeper networks by introducing the idea of residual learning. Instead of widening the network, the increased depth of the network results in fewer additional parameters. This greatly reduces the training time and improves accuracy (highest accuracy ResNet-50: 99.20%; highest accuracy ResNet-101: 99.01%). Mitigates the effect of vanishing gradient.	A complex architecture Many layers may provide very little or no information. Redundant feature-maps may happen to be relearned.
GoogLeNet layer depth: 22 parameters: 7 million	Computational and memory efficiency. Reduced number of parameters by using bottleneck and global average pooling layer. Use of auxiliary classifiers to improve the convergence rate.	Lower accuracy (highest accuracy: 98.77%). Its heterogeneous topology necessitates adaptation from module to module. Substantially reduces the feature space because of the representation bottleneck and thus sometimes may lead to loss of useful information.

**Table 5 sensors-21-05207-t005:** Multi-class classification of LF from Ga dataset for CO (Class 0), PD Stage 2 (Class 2), PD Stage 2.5 (Class 2.5), and PD Stage 3 (Class 3) using several CNN classifiers (AlexNet, ResNet-50, ResNet-101, and GoogLeNet) with 10-fold cross-validation.

Time Window	Frequency Range	Disease Severity (Class)	AlexNet				ResNet-50				ResNet-101				GoogLeNet			
			Sen (%)	Spec (%)	Acc (%)	AUC	Sen (%)	Spec (%)	Acc (%)	AUC	Sen (%)	Spec (%)	Acc (%)	AUC	Sen (%)	Spec (%)	Acc (%)	AUC
10 s	0.83–1.95 Hz	Class 0	98.21	98.11	98.14	0.9816	97.54	99.44	98.81	0.9849	97.54	99.56	98.89	0.9855	97.54	97.89	97.77	0.9771
		Class 2	93.29	97.78	96.14	0.9554	95.53	97.66	96.88	0.9659	97.15	96.49	96.73	0.9682	93.09	97.89	96.14	0.9549
		Class 2.5	82.50	96.12	93.69	0.8931	91.25	95.93	95.10	0.9359	88.75	97.29	95.77	0.9302	94.17	94.67	94.58	0.9442
		Class 3	83.93	97.37	95.69	0.9065	83.33	98.98	97.03	0.9116	88.10	99.32	97.92	0.9371	73.81	99.41	96.21	0.8661
	1.95–50 Hz	Class 0	97.76	98.56	98.29	0.9816	95.30	98.89	97.70	0.9710	94.63	98.44	97.18	0.9654	97.09	97.78	97.55	0.9743
		Class 2	95.93 *	98.25 *	97.40 *	0.9709 *	96.54	96.26	96.36	0.9640	96.34	94.74	95.32	0.9554	93.09	97.66	95.99	0.9538
		Class 2.5	93.75 *	98.83 *	97.92 *	0.9629 *	91.67	98.55	97.33	0.9511	87.50	98.01	96.14	0.9276	90.42	97.65	96.36	0.9403
		Class 3	96.43	99.15	98.81	0.9779	94.64	99.24	98.66	0.9694	89.29	99.24	98	0.9426	92.86	98.64	97.92	0.9575
15 s	0.83–1.95 Hz	Class 0	97.98	98.33	98.22	0.9816	93.94	98.50	96.99	0.9622	95.29	99.33	97.99	0.9731	97.98 *	99.17 *	98.77 *	0.9857 *
		Class 2	94.21	96.66	95.76	0.9543	93.90	95.08	94.65	0.9449	97.26	95.43	96.10	0.9634	96.04	96.84	96.54	0.9644
		Class 2.5	85	97.29	95.09	0.9114	88.75	96.74	95.32	0.9275	90	98.51	96.99	0.9425	88.75	97.69	96.10	0.9322
		Class 3	90.18	98.60	97.55	0.9439	88.39	98.98	97.66	0.9369	95.54	99.62	99.11	0.9758	91.96	99.24	98.33	0.9560
	1.95–50 Hz	Class 0	96.30	98.33	97.66	0.9731	93.94	98.50	96.99	0.9622	95.96	96.33	96.21	0.9615	95.96	98	97.32	0.9698
		Class 2	93.29	97.54	95.99	0.9542	93.90	95.08	94.65	0.9449	91.77	95.78	94.31	0.9378	94.51	95.78	95.32	0.9515
		Class 2.5	88.13	97.56	95.88	0.9284	88.75	96.74	95.32	0.9275	89.38	96.47	95.21	0.9292	84.38	96.74	94.54	0.9056
		Class 3	94.64	97.96	97.55	0.9630	88.39	98.98	97.66	0.9369	81.25	99.36	97.10	0.9031	84.82	98.47	96.77	0.9165
30 s	0.83–1.95 Hz	Class 0	95.24	98.33	97.32	0.9679	95.92	98.33	97.54	0.9713	91.84	99.67	97.09	0.9575	94.56	96.33	95.75	0.9545
		Class 2	93.90	97.17	95.97	0.9554	92.68	95.41	94.41	0.9404	95.73	91.87	93.29	0.9380	90.85	94.70	93.29	0.9278
		Class 2.5	95	97.55	97.09	0.9627	90	95.64	94.63	0.9282	85	94.82	93.06	0.8991	85	97.82	95.53	0.9141
		Class 3	94.64	99.49	98.88	0.9707	83.93	99.74	97.76	0.9184	76.79	99.74	96.87	0.8826	92.86	98.72	97.99	0.9579
	1.95–50 Hz	Class 0	95.92	98	97.32	0.9696	93.20	99	97.09	0.9610	93.20	97.33	95.97	0.9527	95.24	97.33	96.64	0.9629
		Class 2	93.29	95.76	94.85	0.9453	93.90	94.35	94.18	0.9412	93.30	92.58	93.06	0.9324	88.41	94.35	92.17	0.9138
		Class 2.5	80	98.37	95.08	0.8918	86.25	94.82	93.29	0.9054	82.50	95.91	93.51	0.8921	76.25	94.28	91.05	0.8526
		Class 3	98.21 *	97.44 *	97.54 *	0.9783 *	80.36	98.98	96.64	0.8967	76.79	99.23	96.42	0.8801	83.93	97.70	95.97	0.9081

Note: * selected by Youden’s index criteria as the best classification result.

**Table 6 sensors-21-05207-t006:** Multi-class classification of LF from Ju dataset for CO (Class 0), PD Stage 2 (Class 2), PD Stage 2.5 (Class 2.5), and PD Stage 3 (Class 3) using several CNN classifiers (AlexNet, ResNet-50, ResNet-101, and GoogLeNet) with 10-fold cross-validation.

Time Window	Frequency Range	Disease Severity (Class)	AlexNet				ResNet-50				ResNet-101				GoogLeNet			
			Sen (%)	Spec (%)	Acc (%)	AUC	Sen (%)	Spec (%)	Acc (%)	AUC	Sen (%)	Spec (%)	Acc (%)	AUC	Sen (%)	Spec (%)	Acc (%)	AUC
10 s	0.83–1.95 Hz	Class 0	94.47	99.02	98.21	0.9675	96.98 *	99.67 *	99.20 *	0.9833 *	93.97	99.02	98.13	0.9650	97.49	98.48	98.30	0.9798
		Class 2	93.75	98.05	96.70	0.9590	96.59 *	98.44 *	97.86 *	0.9751 *	95.17	94.66	94.82	0.9492	93.75	95.96	95.27	0.9486
		Class 2.5	94.13	94.85	94.55	0.9449	98.26	93.64	95.54	0.9595	92.61	95.61	94.38	0.9411	93.26	94.09	93.75	0.9368
		Class 3	80.73	97.73	96.07	0.8923	70.64	99.90	97.05	0.8527	80.73	99.51	97.68	0.9012	71.56	99.51	96.79	0.8553
	1.95–50 Hz	Class 0	96.48	98.70	98.30	0.9759	94.47	99.46	98.57	0.9696	94.97	98.91	98.21	0.9694	92.46	98.70	97.59	0.9558
		Class 2	92.90	98.70	96.88	0.9580	93.18	97.01	95.80	0.9509	95.45	94.79	95	0.9512	93.47	96.35	95.45	0.9491
		Class 2.5	97.39 *	96.52 *	96.88 *	0.9695 *	96.52	95.45	95.89	0.9599	92.83	96.97	95.27	0.9490	93.70	97.12	95.71	0.9541
		Class 3	90.83	99.11	98.30	0.9497	89.91	99.60	98.66	0.9476	87.16	99.70	98.48	0.9343	93.58	98.52	98.04	0.9605
15 s	0.83–1.95 Hz	Class 0	93.80	99.17	98.22	0.9648	96.12	98.84	98.36	0.9748	93.80	98.67	97.81	0.9623	93.02	99.50	98.36	0.9626
		Class 2	95.63	97.60	96.99	0.9662	91.27	98	95.89	0.9464	92.14	97.41	95.75	0.9477	96.51	95.81	96.03	0.9616
		Class 2.5	94.33	96.98	95.89	0.9566	96.67	95.35	95,89	0.9601	96.67	94.88	95.62	0.9578	92	97.21	95.07	0.9460
		Class 3	88.89	98.02	97.12	0.9346	90.28	99.24	98.36	0.9476	83.33	99.24	97.67	0.9129	90.28	98.18	97.40	0.9423
	1.95–50 Hz	Class 0	90.70	98.50	97.12	0.9460	87.60	99.17	97.12	0.9338	93.80	98.17	97.40	0.9598	89.92	98	96.58	0.9396
		Class 2	93.01	97.01	95.75	0.9501	93.89	96.01	95.34	0.9495	93.45	94.61	94.25	0.9403	88.21	96.41	93.84	0.9231
		Class 2.5	94.67	97.44	96.30	0.9605	94.33	96.05	95.34	0.9519	91.67	95.58	93.97	0.9362	94.67	93.95	94.25	0.9431
		Class 3	94.44	98.02	97.67	0.9623	88.89	98.02	97.12	0.9346	79.17	99.09	97.12	0.8913	86.11	98.48	97.26	0.9230
30 s	0.83–1.95 Hz	Class 0	86.21	98.52	96.37	0.9237	91.38	100	98.49	0.9569	89.66	100	98.19	0.9483	84.48	98.17	95.77	0.9133
		Class 2	92.23	93.86	93.35	0.9305	96.12	96.49	96.37	0.9630	96.12	95.61	95.77	0.9587	85.44	94.74	91.84	0.9009
		Class 2.5	91.85	96.94	94.86	0.9440	96.30	93.88	94.86	0.9509	96.30	89.29	92.15	0.9279	96.30	92.35	93.96	0.9432
		Class 3	97.14 *	98.65 *	98.49 *	0.9790 *	77.14	99.32	96.98	0.8823	51.43	99.66	94.56	0.7555	85.71	99.32	97.89	0.9252
	1.95–50 Hz	Class 0	89.66	97.07	95.77	0.9336	79.31	100	96.37	0.8966	81.03	98.53	95.47	0.8978	82.76	94.87	92.75	0.8882
		Class 2	83.50	96.05	92.15	0.8977	85.44	93.42	90.94	0.8943	90.29	90.79	90.63	0.9054	79.61	92.98	88.82	0.8630
		Class 2.5	93.33	93.37	93.35	0.9335	95.56	87.24	90.63	0.9140	91.11	87.76	89.12	0.8943	89.63	93.37	91.84	0.9150
		Class 3	91.43	98.31	97.58	0.9487	71.43	98.89	96.07	0.8521	48.57	99.32	93.96	0.7395	82.86	97.30	95.77	0.9008

Note: * selected by Youden’s index criteria as the best classification result.

**Table 7 sensors-21-05207-t007:** Multi-class classification of LF from Si dataset for CO (Class 0), PD Stage 2 (Class 2), PD Stage 2.5 (Class 2.5), and PD Stage 3 (Class 3) using several CNN classifiers (AlexNet, ResNet-50, ResNet-101, and GoogLeNet) with 10-fold cross-validation.

Time Window	Frequency Range	Disease Severity (Class)	AlexNet				ResNet-50				ResNet-101				GoogLeNet			
			Sen (%)	Spec (%)	Acc (%)	AUC	Sen (%)	Spec (%)	Acc (%)	AUC	Sen (%)	Spec (%)	Acc (%)	AUC	Sen (%)	Spec (%)	Acc (%)	AUC
10 s	0.83–1.95 Hz	Class 0	97.99	98.10	98.05	0.9804	97.70	99.52	98.70	0.9861	97.99	97.38	97.66	0.9768	97.13	96.67	96.88	0.9690
		Class 2	94.64	91.20	92.71	0.9292	97.92	95.60	96.61	0.9676	95.83	94.91	95.31	0.9537	94.35	92.82	93.49	0.9358
		Class 2.5	61.90	98.39	94.40	0.8015	86.90	99.27	97.92	0.9309	82.14	99.56	97.66	0.9085	73.81	99.12	96.35	0.8647
	1.95–50 Hz	Class 0	98.56 *	99.05 *	98.83 *	0.9881 *	96.84	99.05	98.05	0.9794	96.26	96.43	96.35	0.9635	95.98	98.57	97.40	0.9727
		Class 2	95.83 *	98.84 *	97.53 *	0.9734 *	97.32	96.06	96.61	0.9669	95.54	93.52	94.40	0.9453	94.64	93.98	94.27	0.9431
		Class 2.5	98.81 *	98.39 *	98.44 *	0.9860 *	91.67	99.12	98.31	0.9539	80.95	99.85	97.79	0.9040	84.52	98.10	96.61	0.9131
15 s	0.83–1.95 Hz	Class 0	97.41	97.50	97.46	0.9746	96.12	99.29	97.85	0.9770	96.55	97.86	97.27	0.9720	99.14	97.14	98.05	0.9814
		Class 2	92.41	94.79	93.75	0.9360	96.88	94.10	95.31	0.9549	95.98	95.14	95.51	0.9556	93.30	97.57	95.70	0.9544
		Class 2.5	82.14	97.59	95.90	0.8987	85.71	98.90	97.46	0.9231	87.50	99.12	97.85	0.9331	91.07	98.46	97.66	0.9477
	1.95–50 Hz	Class 0	94.83	98.57	96.88	0.9670	93.53	98.57	96.29	0.9605	96.12	97.50	96.88	0.9681	96.12	96.79	96.48	0.9645
		Class 2	93.30	95.49	94.53	0.9439	96.43	92.36	94.14	0.9439	95.09	93.40	94.14	0.9425	95.54	94.44	94.92	0.9499
		Class 2.5	96.43	97.37	97.27	0.9690	85.71	98.90	97.46	0.9231	80.36	98.90	96.88	0.8963	85.71	99.56	98.05	0.9264
30 s	0.83–1.95 Hz	Class 0	93.97	96.43	95.31	0.9520	93.97	97.86	96.09	0.9591	94.83	96.43	95.70	0.9563	95.69	97.86	96.88	96.77
		Class 2	92.86	93.75	93.36	0.9330	94.64	93.75	94.14	0.9420	95.54	88.19	91.41	0.9187	92.86	93.75	93.36	93.30
		Class 2.5	89.29	98.25	97.27	0.9377	89.29	98.25	97.27	0.9377	57.14	99.56	94.92	0.7835	82.14	97.37	95.70	89.76
	1.95–50 Hz	Class 0	97.41	94.29	95.70	0.9585	93.10	98.57	96.09	0.9584	94.83	95	94.92	0.9491	93.10	98.57	96.09	0.9584
		Class 2	88.39	93.75	91.41	0.9107	93.75	89.58	91.41	0.9167	90.18	89.58	89.84	89.88	91.07	93.06	92.19	0.9206
		Class 2.5	75	97.37	94.92	0.8618	71.43	97.37	94.53	0.8440	64.29	97.81	94.14	81.05	92.86	96.49	96.09	0.9467

Note: * selected by Youden’s index criteria as the best classification result.

**Table 8 sensors-21-05207-t008:** Multi-class classification of RF from Ga dataset for CO (Class 0), PD Stage 2 (Class 2), PD Stage 2.5 (Class 2.5), and PD Stage 3 (Class 3) using several CNN classifiers (AlexNet, ResNet-50, ResNet-101, and GoogLeNet) with 10-fold cross-validation.

Time Window	Frequency Range	Disease Severity (Class)	AlexNet				ResNet-50				ResNet-101				GoogLeNet			
			Sen (%)	Spec (%)	Acc (%)	AUC	Sen (%)	Spec (%)	Acc (%)	AUC	Sen (%)	Spec (%)	Acc (%)	AUC	Sen (%)	Spec (%)	Acc (%)	AUC
10 s	0.83–1.95 Hz	Class 0	98.88	98.11	98.37	0.9850	97.76	99.11	98.66	0.9844	98.43 *	99 *	98.81 *	0.9872 *	96.20	98.89	98	0.9754
		Class 2	95.33	95.56	95.47	0.9544	94.92	97.54	96.59	0.9623	95.53	97.19	96.59	0.9636	93.50	96.61	95.47	0.9505
		Class 2.5	76.25	96.21	92.65	0.8623	91.67	97.02	96.07	0.9434	89.58	95.93	94.80	0.9276	89.17	94.94	93.91	0.9205
		Class 3	78.57	97.96	95.55	0.8827	90.48	99.24	98.14	0.9486	80.95	99.32	97.03	0.9014	79.76	98.81	96.44	0.8929
	1.95–50 Hz	Class 0	97.76	99.67	99.03	0.9871	94.85	98.89	97.55	0.9687	96.20	97.78	97.25	0.9699	96.42	97.56	97.18	0.9699
		Class 2	97.97 *	97.78 *	97.85 *	0.9787 *	96.14	96.02	96.07	0.9608	94.31	96.26	95.55	0.9528	90.04	96.96	94.43	0.9350
		Class 2.5	89.17	98.28	96.66	0.9373	91.67	98.28	97.10	0.9498	90	97.29	95.99	0.9364	86.67	94.85	93.39	0.9076
		Class 3	93.45	98.64	98	0.9605	94.05 *	99.24 *	98.59 *	0.9664 *	86.90	99.24	97.70	0.9307	82.14	98.13	96.14	0.9014
15 s	0.83–1.95 Hz	Class 0	97.31	99	98.44	0.9815	96.63	99.50	98.55	0.9807	95.96	99	97.99	0.9748	98.98	98.17	98.10	0.9807
		Class 2	93.90	98.24	96.66	0.9607	96.95	97.01	96.99	0.9698	96.04	96.13	96.10	0.9609	90.85	96.66	94.54	0.9376
		Class 2.5	96.88 *	96.88 *	96.88 *	0.9688 *	92.50	97.83	96.88	0.9516	92.50	97.15	96.32	0.9483	87.50	95.79	94.31	0.9165
		Class 3	91.96	99.82	98.66	0.9579	92.86	99.49	98.66	0.9617	88.39	99.87	98.44	0.9413	89.29	99.11	97.88	0.9420
	1.95–50 Hz	Class 0	98.32	98.33	98.33	0.9832	95.29	98.50	97.44	0.9689	93.94	98.67	97.10	0.9630	95.96	97.33	96.88	0.9665
		Class 2	94.21	98.77	97.10	0.9649	92.38	97.01	95.32	0.9470	95.43	93.67	94.31	0.9455	91.16	94.73	93.42	0.9294
		Class 2.5	87.50	97.96	96.10	0.9273	90.63	96.61	95.54	0.9362	83.75	95.93	93.76	0.8984	78.13	97.56	94.09	0.8784
		Class 3	94.64	97.71	97.32	0.9617	91.96	98.47	97.66	0.9522	79.46	98.98	96.54	0.8922	95.54	97.83	97.55	0.9669
30 s	0.83–1.95 Hz	Class 0	96.60	98.67	97.99	0.9763	95.24	99.67	98.21	0.9745	94.56	99.33	97.76	0.9695	93.20	98.33	96.64	0.9577
		Class 2	96.34	96.47	96.42	0.9640	96.34	97.17	96.87	0.9676	96.34	92.58	93.96	0.9446	95.73	92.23	93.51	0.9398
		Class 2.5	88.75	98.37	96.64	0.9356	96.25	97	96.87	0.9663	81.25	94.28	91.95	0.8776	77.50	97.28	93.74	0.8739
		Class 3	92.86	98.98	98.21	0.9592	89.29	99.49	98.21	0.9439	69.64	99.49	95.75	0.8457	85.71	98.47	96.87	0.9209
	1.95–50 Hz	Class 0	93.20	97.33	95.97	0.9527	91.84	97.67	95.75	0.9475	93.88	97.67	96.42	0.9577	93.88	96.33	95.53	0.9511
		Class 2	92.07	94.35	93.51	0.9321	88.41	94.35	92.17	0.9138	91.46	92.23	91.95	0.9184	85.98	93.29	90.60	0.8963
		Class 2.5	80	97.55	94.41	0.8877	90	94.01	93.29	0.9200	81.25	92.64	90.60	0.8695	73.75	94.01	90.38	0.8388
		Class 3	92.86	97.44	96.87	0.9515	82.14	98.98	96.87	0.9056	64.29	99.49	95.08	0.8189	82.14	97.19	95.30	0.8966

Note: * selected by Youden’s index criteria as the best classification result.

**Table 9 sensors-21-05207-t009:** Multi-class classification of RF from Ju dataset for CO (Class 0), PD Stage 2 (Class 2), PD Stage 2.5 (Class 2.5), and PD Stage 3 (Class 3) using several CNN classifiers (AlexNet, ResNet-50, ResNet-101, and GoogLeNet) with 10-fold cross-validation.

Time Window	Frequency Range	Disease Severity (Class)	AlexNet				ResNet-50				ResNet-101				GoogLeNet			
			Sen (%)	Spec (%)	Acc (%)	AUC	Sen (%)	Spec (%)	Acc (%)	AUC	Sen (%)	Spec (%)	Acc (%)	AUC	Sen (%)	Spec (%)	Acc (%)	AUC
10 s	0.83–1.95 Hz	Class 0	96.98	99.35	98.93	0.9817	95.98	99.46	98.84	0.9772	95.48	99.57	98.84	0.9752	94.97	99.13	98.39	0.9705
		Class 2	94.89	98.31	97.23	0.9660	97.16	97.79	97.59	0.9747	97.73 *	97.40 *	97.50 *	0.9756 *	96.59	95.70	95.98	0.9615
		Class 2.5	95.43	94.70	95	0.9507	96.30	95.45	95.80	0.9588	95.87 *	96.36 *	96.16 *	0.9612 *	94.13	92.12	92.95	0.9313
		Class 3	78.90	98.62	96.70	0.8876	77.06	99.21	97.05	0.8814	81.65	99.21	97.50	0.9043	56.88	99.70	95.54	0.7829
	1.95–50 Hz	Class 0	97.99 *	99.24 *	99.02 *	0.9861 *	92.96	99.24	98.13	0.9610	93.97	98.81	97.95	0.9639	90.95	98.91	97.50	0.9493
		Class 2	93.47	98.44	96.88	0.9595	91.48	97.14	95.36	0.9431	95.74	94.53	94.91	0.9513	93.75	95.05	94.64	0.9440
		Class 2.5	94.13	96.82	95.71	0.9547	95.65	94.39	94.91	0.9502	89.57	97.88	94.46	0.9372	91.74	96.97	94.82	0.9435
		Class 3	95.41	98.12	97.86	0.9677	87.16	98.81	97.68	0.9298	90.83	98.22	97.50	0.9452	92.66	98.22	97.68	0.9544
15 s	0.83–1.95 Hz	Class 0	96.12	99.17	98.63	0.9765	96.90	99.33	98.90	0.9812	93.02	99.50	98.36	0.9626	89.15	99.17	97.40	0.9416
		Class 2	95.63	97.41	96.85	0.9652	95.63	97.60	96.99	0.9662	95.63	96.41	96.16	09602	95.20	96.21	95.89	0.9570
		Class 2.5	93	96.98	95.34	0.9499	94	97.91	96.30	0.9595	95.67	96.51	96.16	0.9609	94.67	96.28	95.62	0.9547
		Class 3	88.89	98.02	97.12	0.9346	95.83 *	98.48 *	98.22 *	0.9716 *	90.28	99.54	98.63	0.9491	86.11	98.33	97.12	0.9222
	1.95–50 Hz	Class 0	93.02	99.17	98.08	0.9610	88.37	99	97.12	0.9369	90.70	99.67	98.08	0.9518	84.50	97.50	95.21	0.9100
		Class 2	94.32	95.41	95.07	0.9487	93.01	94.81	94.25	0.9391	96.94	94.61	95.34	0.9578	84.72	94.21	91.23	0.8946
		Class 2.5	91.67	96.74	94.66	0.9421	94.33	94.88	94.66	0.9461	93.33	93.72	93.56	0.9353	94.33	92.79	93.42	0.9356
		Class 3	91.67	98.33	97.67	0.9500	83.33	99.09	97.53	0.9121	69.44	99.24	96.30	0.8434	84.72	98.78	97.40	0.9175
30 s	0.83–1.95 Hz	Class 0	86.21	99.63	97.28	0.9292	86.21	99.27	96.98	0.9274	87.93	100	97.89	0.9397	93.10	97.07	96.37	0.9509
		Class 2	96.12	93.42	94.26	0.9477	95.15	93.86	94.26	0.9450	89.32	95.61	93.66	0.9247	87.38	93.86	91.84	0.9062
		Class 2.5	90.37	97.45	94.56	0.9391	92.59	96.43	94.86	0.9451	97.04	86.73	90.94	0.9189	90.37	95.41	93.35	0.9289
		Class 3	94.29	97.97	97.58	0.9613	88.57	98.65	97.58	0.9361	57.14	99.66	95.17	0.7840	88.57	98.99	97.89	0.9378
	1.95–50 Hz	Class 0	84.48	98.17	95.77	0.9133	82.76	98.90	96.07	0.9083	86.21	98.17	96.07	0.9219	82.76	98.17	95.47	0.9046
		Class 2	91.26	93.86	93.05	0.9256	86.41	92.98	90.94	0.8970	91.26	92.54	92.15	0.9190	88.35	92.98	91.54	0.9067
		Class 2.5	94.81	95.41	95.17	0.9511	93.33	91.84	92.45	0.9259	90.37	89.80	90.03	0.9008	87.41	93.88	91.24	0.9064
		Class 3	88.57	99.66	98.49	0.9412	85.71	98.99	97.58	0.9235	54.29	98.65	93.96	0.7647	85.71	96.28	95.17	0.9100

Note: * selected by Youden’s index criteria as the best classification result.

**Table 10 sensors-21-05207-t010:** Multi-class classification of RF from Si dataset for CO (Class 0), PD Stage 2 (Class 2), PD Stage 2.5 (Class 2.5), and PD Stage 3 (Class 3) using several CNN classifiers (AlexNet, ResNet-50, ResNet-101, and GoogLeNet) with 10-fold cross-validation.

Time Window	Frequency Range	Disease Severity (Class)	AlexNet				ResNet-50				ResNet-101				GoogLeNet			
			Sen (%)	Spec (%)	Acc (%)	AUC	Sen (%)	Spec (%)	Acc (%)	AUC	Sen (%)	Spec (%)	Acc (%)	AUC	Sen (%)	Spec (%)	Acc (%)	AUC
10 s	0.83–1.95 Hz	Class 0	97.99	97.86	97.92	0.9792	97.92	97.41	98.33	0.9787	99.43 *	97.86 *	98.57 *	0.9864 *	96.26	97.14	96.74	0.9670
		Class 2	95.24	94.21	94.66	0.9473	95.96	95.54	96.30	0.9592	95.83 *	97.22 *	96.61 *	0.9653 *	95.24	89.58	92.06	0.9241
		Class 2.5	77.38	98.83	96.48	0.8811	97.79	90.48	98.68	0.9458	88.10	99.27	98.05	0.9368	59.52	99.12	94.79	0.7932
	1.95–50 Hz	Class 0	98.28	97.38	97.79	0.9783	94.25	99.29	97.01	0.9677	97.99	96.19	97.01	0.9709	96.84	96.90	96.88	0.9687
		Class 2	92.26	97.45	95.18	0.9486	98.21	93.75	95.70	0.9598	95.24	95.83	95.57	0.9554	88.10	96.06	92.58	0.9208
		Class 2.5	92.86	97.66	97.14	0.9526	90.48	99.42	98.44	0.9495	85.71	99.85	98.31	0.9278	91.67	95.91	95.44	0.9379
15 s	0.83–1.95 Hz	Class 0	97.41	97.86	97.66	0.9764	98.28	97.86	98.05	0.9807	95.26	98.57	97.07	0.9692	98.28	97.14	97.66	0.9771
		Class 2	92.41	92.71	92.58	0.9256	95.09	96.53	95.90	0.9581	96.88	93.40	94.92	0.9514	92.86	95.49	94.34	0.9417
		Class 2.5	71.43	97.37	94.53	0.8440	89.29	98.90	97.85	0.9409	85.71	99.34	97.85	0.9253	82.14	98.03	96.29	0.9008
	1.95–50 Hz	Class 0	96.98	96.07	96.48	0.9653	93.53	97.86	95.90	0.9570	96.55	94.64	95.51	0.9560	94.40	95.36	94.92	0.9488
		Class 2	93.30	96.53	95.12	0.9492	93.75	94.10	93.95	0.9392	90.63	94.44	92.77	0.9253	90.18	94.44	92.58	0.9231
		Class 2.5	92.86	98.90	98.24	0.9588	94.64 *	98.03 *	97.66 *	0.9633 *	83.93	98.46	96.88	0.9120	92.86	97.81	97.27	0.9533
30 s	0.83–1.95 Hz	Class 0	89.66	99.29	94.92	0.9447	93.10	97.86	95.70	0.9548	93.10	95	94.14	0.9405	91.38	92.86	92.19	0.9212
		Class 2	97.32	88.19	92.19	0.9276	96.43	90.97	93.36	0.9370	91.07	90.28	90.63	0.9067	86.61	92.36	89.84	0.8948
		Class 2.5	78.57	98.68	96.48	0.8863	78.57	99.12	96.88	0.8885	75	98.25	95.70	0.8662	92.86	97.37	96.88	0.9511
	1.95–50 Hz	Class 0	96.55	95.71	96.09	0.9613	90.52	97.14	94.14	0.9383	95.69	94.29	94.92	0.9499	92.24	97.14	94.92	0.9469
		Class 2	88.39	93.75	91.41	0.9107	91.96	89.58	90.63	0.9077	91.96	87.50	89.45	0.8973	88.39	93.75	91.41	0.9107
		Class 2.5	78.57	96.49	94.53	0.8753	82.14	97.37	95.70	0.8976	50	99.12	93.75	0.7456	96.43	95.61	95.70	0.9602

Note: * selected by Youden’s index criteria as the best classification result.

**Table 11 sensors-21-05207-t011:** Multi-class classification of CF Ga Dataset for CO (Class 0), PD Stage 2 (Class 2), PD Stage 2.5 (Class 2.5), and PD Stage 3 (Class 3) using several CNN classifiers (AlexNet, ResNet-50, ResNet-101, and GoogLeNet) with 10-fold cross-validation.

Time Window	Frequency Range	Disease Severity (Class)	AlexNet				ResNet-50				ResNet-101				GoogLeNet			
			Sen (%)	Spec (%)	Acc (%)	AUC	Sen (%)	Spec (%)	Acc (%)	AUC	Sen (%)	Spec (%)	Acc (%)	AUC	Sen (%)	Spec (%)	Acc (%)	AUC
10 s	0.83–1.95 Hz	Class 0	96.42	98.56	97.85	0.9749	97.32	99.44	98.74	0.9838	96.87	99.44	98.59	0.9816	96.20	99.33	98.29	0.9777
		Class 2	94.51	95.91	95.40	0.9521	94.11	97.54	96.29	0.9582	97.15	95.67	96.21	0.9641	94.51	95.56	95.17	0.9503
		Class 2.5	90.42	93.95	93.32	0.9218	92.92	96.21	95.62	0.9456	87.92	96.48	94.95	0.9220	81.67	94.94	92.58	0.8830
		Class 3	68.45	99.66	95.77	0.8406	89.29	99.32	98.07	0.9430	82.14	99.49	97.33	0.9082	80.36	98.22	95.99	08929
	1.95–50 Hz	Class 0	97.99	99.44	98.96	0.9872	95.53	98.78	97.70	0.9715	96.87	97.78	97.48	0.9732	96.87	99.22	98.44	0.9805
		Class 2	96.54	98.48	97.77	0.9751	94.92	96.02	95.62	0.9547	94.51	95.56	95.17	0.9503	94.51	96.84	95.99	0.9568
		Class 2.5	92.50	98.28	97.25	0.9539	90.42	97.56	96.29	0.9399	85	98.46	96.07	0.9173	92.08	96.75	95.92	0.9442
		Class 3	95.83	98.81	98.44	0.9732	92.26	99.24	98.37	0.9575	94.05	98.98	98.37	0.9651	90.48	99.49	98.37	0.9498
15 s	0.83–1.95 Hz	Class 0	95.96	97.67	97.10	0.9681	97.64	99.17	98.66	0.9840	96.63	99.17	98.33	0.9790	92.59	98.83	96.77	0.9571
		Class 2	92.99	96.31	95.09	0.9465	92.99	98.42	96.43	0.9570	93.60	97.01	95.76	0.9530	92.68	94.90	94.09	0.9379
		Class 2.5	87.50	96.34	94.76	0.9192	90	96.07	94.98	0.9303	88.75	94.44	93.42	0.9159	83.13	94.30	92.31	0.8871
		Class 3	83.93	98.60	96.77	0.9126	91.07	98.34	97.44	0.9471	77.68	98.60	95.99	0.8814	77.68	97.45	94.98	0.8757
	1.95–50 Hz	Class 0	98.65 *	99 *	98.89 *	0.9883 *	97.31	99.17	98.55	0.9824	96.30	97.50	97.10	0.9690	96.63	99.17	98.33	0.9790
		Class 2	96.65 *	98.42 *	97.77 *	0.9753 *	96.04	97.54	96.99	0.9679	94.82	94.38	94.54	0.9460	95.43	97.72	96.88	0.9657
		Class 2.5	91.88 *	99.19 *	97.88 *	0.9553 *	92.50	97.69	96.77	0.9510	83.13	97.29	94.76	0.9021	90	96.20	95.09	0.9310
		Class 3	99.11 *	98.98 *	99 *	0.9904 *	91.96	99.24	98.33	0.9560	83.93	90.24	97.32	0.9158	83.93	98.34	96.54	0.9114
30 s	0.83–1.95 Hz	Class 0	91.16	96.67	94.85	0.9391	93.20	97.67	96.20	0.9543	91.84	99.33	96.87	0.9559	91.84	97.33	95.53	0.9459
		Class 2	87.20	90.81	89.49	0.8900	90.85	93.99	92.84	0.9242	94.51	92.58	93.29	0.9355	89.02	93.64	91.95	0.9133
		Class 2.5	70	95.10	90.60	0.8255	87.50	95.10	93.74	0.9130	82.50	95.10	92.84	0.8880	85	95.10	93.29	0.9005
		Class 3	87.50	97.19	95.97	0.9234	82.14	99.23	97.09	0.9069	80.36	98.72	96.42	0.8954	89.29	98.98	97.76	0.9413
	1.95–50 Hz	Class 0	95.24	97.67	96.87	0.9645	92.52	98.33	96.42	0.9543	93.88	97.33	96.20	0.9561	95.92	94.67	95.08	0.9529
		Class 2	91.46	96.11	94.41	0.9379	93.29	94.70	94.18	0.9400	94.51	93.64	93.96	0.9408	84.76	96.47	92.17	0.9061
		Class 2.5	85	96.73	94.63	0.9087	85	95.91	93.96	0.9046	87.50	94.82	93.51	0.9116	91.25	95.10	94.41	0.9317
		Class 3	91.07	97.95	97.09	0.9451	83.93	97.95	96.20	0.9094	67.86	99.74	95.75	0.8380	83.93	99.23	97.32	0.9158

Note: * selected by Youden’s index criteria as the best classification result.

**Table 12 sensors-21-05207-t012:** Multi-class classification of CF from Ju dataset for CO (Class 0), PD Stage 2 (Class 2), PD Stage 2.5 (Class 2.5), and PD Stage 3 (Class 3) using several CNN classifiers (AlexNet, ResNet-50, ResNet-101, and GoogLeNet) with 10-fold cross-validation.

Time Window	Frequency Range	Disease Severity (Class)	AlexNet				ResNet-50				ResNet-101				GoogLeNet			
			Sen (%)	Spec (%)	Acc (%)	AUC	Sen (%)	Spec (%)	Acc (%)	AUC	Sen (%)	Spec (%)	Acc (%)	AUC	Sen (%)	Spec (%)	Acc (%)	AUC
10 s	0.83–1.95 Hz	Class 0	93.97	99.02	98.13	0.9650	96.48	99.57	99.02	0.9802	95.48	99.89	99.11	0.9768	94.47	99.57	98.66	0.9702
		Class 2	92.05	97.79	95.98	0.9492	96.02 *	98.18 *	97.50 *	0.9710 *	94.60	98.31	97.14	0.9645	94.03	97.14	96.16	0.9558
		Class 2.5	96.09	93.18	94.38	0.9463	97.39	96.21	96.70	0.9680	98.26	93.33	95.36	0.9580	92.61	93.18	92.95	0.9290
		Class 3	77.06	98.81	96.70	0.8794	86.24	99.51	98.21	0.9287	77.06	99.70	97.50	0.8838	74.31	97.73	95.45	0.8602
	1.95–50 Hz	Class 0	98.49 *	98.91 *	98.84 *	0.9870 *	93.47	99.02	98.04	0.9625	96.98	98.26	98.04	0.9762	91.96	98.70	97.50	0.9533
		Class 2	93.18	98.83	97.05	0.9600	94.32	96.22	95.63	0.9527	93.47	95.70	95	0.9458	92.33	96.48	95.18	0.9441
		Class 2.5	96.52 *	97.88 *	97.32 *	0.9720 *	94.35	97.12	95.98	0.9573	93.48	97.12	95.63	0.9530	96.09	96.52	96.34	0.9630
		Class 3	100 *	99.01 *	99.11 *	0.9951 *	92.66	99.01	98.39	0.9584	88.99	99.70	98.66	0.9435	93.58	99.41	98.84	0.9649
15 s	0.83–1.95 Hz	Class 0	96.12	98.67	98.22	0.9740	95.35	99.33	96.83	0.9734	92.25	99.50	98.22	0.9587	92.25	98.84	97.67	0.9554
		Class 2	93.89	96.21	95.48	0.9505	94.32	97.41	96.44	0.9586	94.32	97.21	96.30	0.9576	93.01	97.21	95.89	0.9511
		Class 2.5	92	96.28	94.52	0.9414	93.67	94.65	94.25	0.9416	96.67	92.79	94.38	0.9473	93.33	94.42	93.97	0.9388
		Class 3	87.50	98.63	97.53	0.9307	80.56	98.18	96.44	0.8937	70.83	99.09	96.30	0.8496	79.17	97.57	95.75	0.8837
	1.95–50 Hz	Class 0	95.35	98.34	97.81	0.9684	88.37	99.50	97.53	0.9394	93.02	98.50	97.53	0.9576	86.82	97	95.21	0.9191
		Class 2	92.58	98	96.30	0.9529	92.58	95.01	94.25	0.9379	92.58	94.21	93.70	0.9339	86.90	94.61	92.19	0.9076
		Class 2.5	94	97.44	96.03	0.9572	94.33	95.35	94.93	0.9484	92.33	93.02	92.74	0.9268	93.33	95.81	94.79	0.9457
		Class 3	94.44	97.87	97.53	0.9616	91.67	98.94	98.22	0.9530	69.44	99.54	96.58	0.8449	91.67	98.48	97.81	0.9507
30 s	0.83–1.95 Hz	Class 0	89.66	98.53	96.98	0.9409	81.03	100	96.68	0.9052	70.69	100	94.86	0.8534	93.10	99.27	98.19	0.9619
		Class 2	89.32	93.42	92.15	0.9137	96.12	92.11	93.35	0.9411	92.23	91.67	91.84	0.9195	92.23	95.18	94.26	0.9370
		Class 2.5	86.67	94.39	91.24	0.9053	90.37	95.92	93.66	0.9314	96.30	89.29	92.15	0.9279	86.67	94.90	91.54	0.9078
		Class 3	88.57	96.96	96.07	0.9277	91.43	98.31	97.58	0.9487	68.57	99.66	96.37	0.8412	88.57	96.28	95.47	0.9243
	1.95–50 Hz	Class 0	91.38	98.17	96.98	0.9477	84.48	100	97.28	0.9224	94.83	98.90	98.19	0.9686	93.10	99.27	98.19	0.9619
		Class 2	93.20	96.05	95.17	0.9463	93.20	95.18	94.56	0.9419	95.15	91.67	92.75	0.9341	95.15	94.74	94.86	0.9494
		Class 2.5	92.59	96.43	94.86	0.9451	93.33	93.88	93.66	0.9361	85.93	92.35	89.73	0.8914	89.63	94.39	92.45	0.9201
		Class 3	85.71	97.97	96.68	0.9184	85.71	97.64	96.37	0.9167	62.86	98.99	95.17	0.8092	80	98.31	96.37	0.8916

Note: * selected by Youden’s index criteria as the best classification result.

**Table 13 sensors-21-05207-t013:** Multi-class classification of CF from Si dataset for CO (Class 0), PD Stage 2 (Class 2), PD Stage 2.5 (Class 2.5), and PD Stage 3 (Class 3) using several CNN classifiers (AlexNet, ResNet-50, ResNet-101, and GoogLeNet) with 10-fold cross-validation.

Time Window	Frequency Range	Disease Severity (Class)	AlexNet				ResNet-50				ResNet-101				GoogLeNet			
			Sen (%)	Spec (%)	Acc (%)	AUC	Sen (%)	Spec (%)	Acc (%)	AUC	Sen (%)	Spec (%)	Acc (%)	AUC	Sen (%)	Spec (%)	Acc (%)	AUC
10 s	0.83–1.95 Hz	Class 0	97.41	95.71	96.48	0.9656	96.84	97.62	97.27	0.9723	97.13	98.57	97.92	0.9785	93.97	99.29	96.88	0.9663
		Class 2	92.56	94.68	93.75	0.9362	94.94	95.37	95.18	0.9516	97.02	93.52	95.05	0.9527	97.02	90.05	93.10	0.9354
		Class 2.5	82.14	98.83	97.01	0.9049	89.29	98.98	97.92	0.9413	78.57	99.42	97.14	0.8899	72.62	98.83	95.96	0.8572
	1.95–50 Hz	Class 0	97.41 *	99.52 *	98.57 *	0.9847 *	95.98	97.62	96.88	0.9680	99.14	95.24	97.01	0.9719	97.13	97.14	97.14	0.9713
		Class 2	94.94 *	97.69 *	96.48 *	0.9631 *	95.24	96.06	95.70	0.9565	93.45	97.45	95.70	0.9545	93.15	96.06	94.79	0.9461
		Class 2.5	97.62	97.66	97.66	0.9764	95.24	98.98	98.57	0.9711	89.29	99.56	98.44	0.9442	90.48	98.25	97.40	0.9436
15 s	0.83–1.95 Hz	Class 0	94.83	95.71	95.31	0.9527	97.41	97.50	97.46	0.9746	94.83	97.50	96.29	0.9616	93.97	98.57	96.48	0.9627
		Class 2	93.30	93.75	93.55	0.9353	92.86	96.53	94.92	0.9469	91.96	94.10	93.16	0.9303	94.64	91.67	92.97	0.9315
		Class 2.5	89.29	99.34	98.24	0.9431	92.86	98.03	97.46	0.9544	91.07	97.59	96.88	0.9433	80.36	98.03	96.09	0.8919
	1.95–50 Hz	Class 0	99.14	95.71	97.27	0.9743	94.83	96.43	95.70	0.9563	96.12	95	95.51	0.9556	94.83	99.29	97.27	0.9706
		Class 2	89.73	99.31	95.12	0.9452	91.96	95.14	93.75	0.9355	93.30	92.71	92.97	0.9301	96.88	92.36	94.34	0.9462
		Class 2.5	98.21 *	97.37 *	97.46 *	0.9779 *	94.64	98.03	97.66	0.9633	76.79	99.56	97.07	0.8817	80.36	98.68	96.68	0.8952
30 s	0.83–1.95 Hz	Class 0	90.52	95.71	93.36	0.9312	96.55	97.86	97.27	0.9720	93.97	100	97.27	0.9698	90.52	93.57	92.19	0.9204
		Class 2	92.86	89.58	91.02	0.9122	93.75	92.36	92.97	0.9306	97.32	91.67	94.14	0.9449	89.29	90.28	89.84	0.8978
		Class 2.5	82.14	98.68	96.88	0.9041	75	98.25	95.70	0.8662	82.14	98.68	96.88	0.9041	85.71	98.25	96.88	0.9198
	1.95–50 Hz	Class 0	93.10	97.86	95.70	0.9548	93.97	97.86	96.09	0.9591	95.69	96.43	96.09	0.9606	91.38	93.57	92.58	0.9248
		Class 2	93.75	93.75	93.75	0.9375	92.86	91.67	92.19	0.9226	95.54	90.28	92.58	0.9291	90.18	90.28	90.23	0.9023
		Class 2.5	92.86	97.81	97.27	0.9533	78.57	97.37	95.31	0.8797	64.29	99.56	95.70	0.8192	82.14	98.68	96.88	0.9041

Note: * selected by Youden’s index criteria as the best classification result.

**Table 14 sensors-21-05207-t014:** Multi-class classification of LF from all datasets for CO (Class 0), PD Stage 2 (Class 2), PD Stage 2.5 (Class 2.5), and PD Stage 3 (Class 3) using several CNN classifiers (AlexNet, ResNet-50, ResNet-101, and GoogLeNet) with 10-fold cross-validation.

Time Window	Frequency Range	Disease Severity (Class)	AlexNet				ResNet-50				ResNet-101				GoogLeNet			
			Sen (%)	Spec (%)	Acc (%)	AUC	Sen (%)	Spec (%)	Acc (%)	AUC	Sen (%)	Spec (%)	Acc (%)	AUC	Sen (%)	Spec (%)	Acc (%)	AUC
10 s	0.83–1.95 Hz	Class 0	92.15	94.56	93.82	0.9335	93.86	96.88	95.95	0.9537	94.27 *	97.68 *	96.63 *	0.9597 *	94.37	95.58	95.21	0.9497
		Class 2	81.61	89.49	86.62	0.8555	84.92	95.09	91.38	0.9000	87.63 *	94.06 *	91.72 *	0.9085 *	76.27	90.51	85.32	0.8339
		Class 2.5	77.81	92.37	88.84	0.8509	91.45 *	93.35 *	92.89 *	0.9240 *	89.16	93.96	92.80	0.9156	79.21	88.82	86.49	0.8402
		Class 3	66.79	98.78	96.04	0.8278	80.14	99.09	97.47	0.8962	80.51	99.32	97.71	0.8991	66.43	99.19	96.38	0.8281
	1.95–50 Hz	Class 0	88.63	93.62	92.09	0.9113	90.14	94.20	92.95	92.17	80.68	96.12	91.38	0.8840	85.61	93.40	91	0.8950
		Class 2	75.25	86.86	82.63	0.8106	75.59	87.45	83.12	81.52	79.49	82.29	81.27	0.8089	74.24	82.77	79.66	0.7851
		Class 2.5	75.64	92.53	88.44	0.8409	75.38	92.17	88.10	83.77	76.15	91.64	87.88	0.8389	66.20	92.86	86.40	0.7953
		Class 3	87.73	98.85	97.90	0.9329	87.73	98.88	97.93	93.31	82.31	99.53	98.05	0.9092	93.14 *	98.17 *	97.74 *	0.9566 *
15 s	0.83–1.95 Hz	Class 0	86.02	94.46	91.87	0.9024	92.40	95.34	94.44	0.9387	91.19	96.42	94.81	0.9380	86.02	94.60	91.96	0.9031
		Class 2	78.87	88.81	85.18	0.8384	84.25	92.78	89.67	0.8852	87.96	91.46	90.18	0.8971	76.70	88	83.87	0.8235
		Class 2.5	83.53	90.33	88.69	0.8693	86.05	95.13	92.94	0.9059	85.47	94.45	92.29	0.8996	76.16	91.44	87.75	0.8380
		Class 3	60.33	98.77	95.47	0.7955	86.41	98.77	97.71	0.9259	73.37	99.13	96.91	0.8625	81.52	97.49	96.12	0.8951
	1.95–50 Hz	Class 0	86.02	93.11	90.93	0.8957	84.50	95.81	92.33	0.9016	89.21	94.80	93.08	0.9201	88.91	90.34	89.90	0.8963
		Class 2	64.40	86.60	78.49	0.7550	77.98	83.58	81.53	0.8078	79.51	81.96	81.07	0.8074	68.76	82.70	77.81	0.7573
		Class 2.5	76.74	87.31	84.76	0.8203	70.93	91.19	86.30	0.8106	64.92	91.37	84.99	0.7815	62.98	91.62	84.71	0.7730
		Class 3	84.24	98.52	97.29	0.9138	82.07	98.52	97.10	0.9029	69.02	99.64	97.01	0.8433	80.98	98.52	97.01	0.8975
30 s	0.83–1.95 Hz	Class 0	83.80	96.91	92.84	0.9036	89.41	96.63	94.39	0.9302	86.92	96.49	93.52	0.9170	82.87	94.81	91.10	0.8884
		Class 2	80.47	83.97	82.69	0.8222	84.70	86.72	85.98	0.8571	86.28	85.65	85.88	0.8596	80.47	83.05	82.11	0.8176
		Class 2.5	64.20	91.40	85.01	0.7780	76.54	92.54	88.78	0.8454	76.13	91.91	88.20	0.8402	67.49	91.91	86.17	0.7970
		Class 3	83.52	96.50	95.36	0.9001	72.53	99.58	97.20	0.8605	59.34	99.36	95.84	0.7935	72.53	97.77	95.55	0.8515
	1.95–50 Hz	Class 0	75.70	94.39	88.59	0.8505	83.18	94.25	90.81	0.8871	79.44	94.95	90.14	0.8720	83.18	88.50	86.85	0.8584
		Class 2	68.60	78.32	74.76	0.7346	76.52	80.15	78.82	0.7833	70.18	80.15	76.50	0.7517	58.31	80.61	72.44	0.6946
		Class 2.5	62.55	87.74	81.82	0.7514	61.32	89.25	82.69	0.7529	67.49	86.22	81.82	0.7685	60.08	86.85	80.56	0.7347
		Class 3	82.42	97.35	96.03	0.8988	57.14	97.88	94.29	0.7751	63.74	98.30	95.26	0.8102	71.43	97.67	95.36	0.8455

Note: * selected by Youden’s index criteria as the best classification result.

**Table 15 sensors-21-05207-t015:** Multi-class classification of RF from all datasets for CO (Class 0), PD Stage 2 (Class 2), PD Stage 2.5 (Class 2.5), and PD Stage 3 (Class 3) using several CNN classifiers (AlexNet, ResNet-50, ResNet-101, and GoogLeNet) with 10-fold cross-validation.

Time Window	Frequency Range	Disease Severity (Class)	AlexNet				ResNet-50				ResNet-101				GoogLeNet			
			Sen (%)	Spec (%)	Acc (%)	AUC	Sen (%)	Spec (%)	Acc (%)	AUC	Sen (%)	Spec (%)	Acc (%)	AUC	Sen (%)	Spec (%)	Acc (%)	AUC
10 s	0.83–1.95 Hz	Class 0	92.05	94.42	93.69	0.9324	94.97 *	96.34 *	95.92 *	0.9566 *	93.66	97.59	96.38	0.9563	90.64	95.31	93.88	0.9298
		Class 2	84.58	90.80	88.53	0.8769	83.98	95.04	91	0.8951	87.88 *	93.72 *	91.59 *	0.9080 *	71.69	89.88	83.25	0.8079
		Class 2.5	82.40	94.57	91.62	0.8849	89.67	93.06	92.24	0.9137	88.65	93.64	92.43	0.9114	80.74	87.03	85.50	0.8388
		Class 3	72.56	99.05	96.79	0.8581	77.26	99.02	97.16	0.8814	75.81	99.22	97.22	0.8752	68.23	98.82	96.20	0.8352
	1.95–50 Hz	Class 0	89.64	92.59	91.68	0.9112	89.54	95.09	93.38	0.9231	88.73	93.98	92.36	0.9135	82.90	93.84	90.48	88.37
		Class 2	67.12	90.27	81.82	0.7869	80.34	86.47	84.23	0.8341	73.90	85.11	81.02	0.7950	70.34	82.82	78.27	76.58
		Class 2.5	77.42	90.37	87.23	0.8390	74.49	93.51	88.90	0.8400	71.17	91.19	86.34	0.8118	73.21	88.98	85.16	81.10
		Class 3	94.95 *	97.30 *	97.09 *	0.9612 *	86.64	99.12	98.05	0.9288	84.48	98.92	97.68	0.9170	79.42	99.12	97.43	89.27
15 s	0.83–1.95 Hz	Class 0	91.79	94.73	93.83	0.9326	92.40	96.49	95.23	0.9445	90.12	97.84	95.47	0.9398	90.58	95.41	93.92	0.9299
		Class 2	80.54	91.02	87.19	0.8578	85.53	94.18	91.02	0.8986	91.04	90.43	90.65	0.9073	78.36	90.65	86.16	0.8450
		Class 2.5	82.95	92.36	90.09	0.8765	89.53 *	94.82 *	93.55 *	0.9218 *	84.88	94.89	92.47	0.8988	79.65	91.99	89.01	0.8582
		Class 3	73.37	99.03	96.82	0.8620	87.50	98.72	97.76	0.9311	75.54	99.34	97.29	0.8744	83.15	97.85	96.59	0.9050
	1.95–50 Hz	Class 0	83.28	93.65	90.46	0.8847	85.26	96.69	93.17	0.9097	86.63	94.87	92.33	0.9075	82.37	92.51	89.39	0.8744
		Class 2	70.81	84.68	79.62	0.7775	79.39	84.46	82.61	0.8192	75.42	85.35	81.72	0.8038	69.27	80.71	76.53	0.7499
		Class 2.5	75.19	89.83	86.30	0.8251	70.93	91.07	86.21	0.8100	75	90.51	86.77	0.8276	64.53	89.77	83.68	0.7715
		Class 3	83.15	98.47	97.15	0.9081	82.07	98.16	96.77	0.9011	79.35	99.08	97.38	0.8921	79.89	98.11	96.54	0.8900
30 s	0.83–1.95 Hz	Class 0	90.03	93.97	92.75	0.9200	90.97	96.21	94.58	0.9359	89.10	95.51	93.52	0.9230	78.50	94.95	89.85	0.8673
		Class 2	77.31	89.62	85.11	0.8346	84.17	91.45	88.78	0.8781	81.79	90.08	87.04	0.8594	77.31	83.05	80.95	0.8018
		Class 2.5	76.54	93.17	89.26	0.8486	86.01	93.30	91.59	0.8965	86.01	90.77	89.65	0.8839	77.78	90.39	87.43	0.8408
		Class 3	87.91	97.77	96.91	0.9284	78.02	99.26	97.39	0.8864	60.44	99.58	96.13	0.8001	71.43	98.73	96.32	0.8508
	1.95–50 Hz	Class 0	83.18	90.60	88.30	0.8689	86.92	92.57	90.81	0.8974	81.62	93.83	90.04	0.8772	80.69	92.57	88.88	0.8663
		Class 2	53.56	84.43	73.11	0.6899	67.55	82.60	77.08	0.7507	64.91	82.60	76.11	0.7375	65.44	80.31	74.85	0.7287
		Class 2.5	68.31	83.06	79.59	0.7569	59.67	87.10	80.66	0.7339	74.07	83.44	81.24	0.7876	60.91	86.09	80.17	0.7350
		Class 3	72.53	96.92	94.78	0.8473	64.84	97.24	94.39	0.8104	53.85	99.15	95.16	0.7650	65.93	97.14	94.39	0.8154

Note: * selected by Youden’s index criteria as the best classification result.

**Table 16 sensors-21-05207-t016:** Multi-class classification of CF from all datasets for CO (Class 0), PD Stage 2 (Class 2), PD Stage 2.5 (Class 2.5), and PD Stage 3 (Class 3) using several CNN classifiers (AlexNet, ResNet-50, ResNet-101, and GoogLeNet) with 10-fold cross-validation.

Time Window	Frequency Range	Disease Severity (Class)	AlexNet				ResNet-50				ResNet-101				GoogLeNet			
			Sen (%)	Spec (%)	Acc (%)	AUC	Sen (%)	Spec (%)	Acc (%)	AUC	Sen (%)	Spec (%)	Acc (%)	AUC	Sen (%)	Spec (%)	Acc (%)	AUC
10 s	0.83–1.95 Hz	Class 0	88.43	95.81	93.54	0.9212	91.65 *	96.21 *	94.81 *	0.9393 *	89.34	97.41	94.93	0.9337	88.73	95.27	93.26	0.9200
		Class 2	78.81	90.46	86.21	0.8464	87.37 *	91.19 *	89.80 *	0.8928 *	88.81	89.29	89.12	0.8905	78.39	89.59	85.50	0.8399
		Class 2.5	85.97	91.19	89.92	0.8858	85.20 *	95.55 *	93.04 *	0.9038 *	83.67	94.53	91.90	0.8910	81.38	91.15	88.78	0.8626
		Class 3	78.34	99.02	97.25	0.8868	83.75	99.39	98.05	0.9157	78.34	99.53	97.71	0.8893	75.45	98.51	96.54	0.8698
	1.95–50 Hz	Class 0	87.83	95.31	93.01	0.9157	88.73	96.56	94.16	0.9265	90.44	96.39	94.56	0.9341	90.44	95.72	94.10	0.9308
		Class 2	83.98	85.26	84.79	0.8462	82.29	87.35	85.50	0.8482	80.34	88.47	85.50	0.8440	73.73	88.61	83.18	0.8117
		Class 2.5	71.56	95.68	89.83	0.8362	78.06	93.15	89.49	0.8560	79.08	91.84	88.75	0.8546	76.28	90.09	86.74	0.8318
		Class 3	92.06 *	98.61 *	98.05 *	0.9534 *	87	99.19	98.15	0.9310	82.31	99.26	97.81	0.9078	89.53	98.41	97.65	0.9397
15 s	0.83–1.95 Hz	Class 0	88.15	95.34	93.13	0.9174	91.03	95.61	94.20	0.9332	86.17	97.16	93.78	0.9167	86.32	94.46	91.96	0.9039
		Class 2	82.59	87.92	85.97	0.8525	82.71	91.68	88.41	0.8720	85.92	88.51	87.56	0.8732	78.87	86.60	83.78	0.8274
		Class 2.5	81.59	93.35	90.51	0.8747	85.66	93.96	91.96	0.8981	83.14	93.96	91.35	0.8855	76.16	92.24	88.36	0.8420
		Class 3	77.72	99.54	97.66	0.8863	84.78	98.98	97.76	0.9188	83.70	98.87	97.57	0.9129	76.63	98.41	96.54	0.8752
	1.95–50 Hz	Class 0	89.36	93.79	92.43	0.9157	85.11	95.75	92.47	0.9043	86.47	96.49	93.41	0.9148	86.93	94.06	91.87	0.9049
		Class 2	78.75	83.43	81.72	0.8109	78.75	86.60	83.73	0.8267	78.75	86.38	83.59	0.8256	70.04	87.04	80.83	0.7854
		Class 2.5	64.15	94.15	86.91	0.7915	78.68	92.05	88.83	0.8537	78.88	90.20	87.47	0.8454	77.52	88.91	86.16	0.8321
		Class 3	89.13	98.52	97.71	0.9382	85.87	98.67	97.57	0.9227	76.63	99.44	97.48	0.8803	81.52	98.67	97.19	0.9010
30 s	0.83–1.95 Hz	Class 0	82.24	94.39	90.62	0.8832	87.54	96.91	94	0.9223	81.93	97.34	92.55	0.8963	85.36	91.87	89.85	0.8861
		Class 2	75.46	82.75	80.08	0.7910	84.70	88.09	86.85	0.8639	81.79	86.87	85.01	0.8433	74.41	82.75	79.69	0.7858
		Class 2.5	67.90	91.28	85.78	0.7959	79.42	92.41	89.36	0.8592	82.72	90.90	88.97	0.8681	63.79	93.68	86.65	0.7873
		Class 3	82.42	97.67	96.32	0.9004	73.63	98.73	96.52	0.8618	75.82	98.52	96.52	0.8717	85.71	97.45	96.42	0.9158
	1.95–50 Hz	Class 0	83.18	92.01	89.26	0.8759	83.19	93.97	90.62	0.8857	84.42	92.29	89.85	0.8835	80.05	91.30	87.81	0.8568
		Class 2	67.81	79.54	75.24	0.7368	71.24	82.60	78.43	0.7692	59.10	87.02	76.79	0.7306	57.52	81.22	72.53	0.6937
		Class 2.5	56.79	89.63	81.91	0.7321	67.90	89.38	84.33	0.7864	78.60	83.69	82.50	0.8115	68.31	84.07	80.37	0.7619
		Class 3	82.42	97.45	96.13	0.8994	79.12	97.99	96.32	0.8855	70.33	98.41	95.94	0.8437	72.53	98.30	96.03	0.8542

Note: * selected by Youden’s index criteria as the best classification result.

**Table 17 sensors-21-05207-t017:** Two-class classification of LF for CO and PD (Class 2, Class 2.5, and Class 3) using several CNN classifiers (AlexNet, ResNet-50, ResNet-101, and GoogLeNet) with 10-fold cross-validation.

Time Window	Frequency Range	Dataset	AlexNet				ResNet-50				ResNet-101				GoogLeNet			
			Sen (%)	Spec (%)	Acc (%)	AUC	Sen (%)	Spec (%)	Acc (%)	AUC	Sen (%)	Spec (%)	Acc (%)	AUC	Sen (%)	Spec (%)	Acc (%)	AUC
10 s	0.83–1.95 Hz	Ga	96.02	99.12	97.92	0.9964	98.87	98.46	98.59	0.9971	98.90	99.02	98.96	0.9996	95.95	98.63	97.62	0.9938
		Ju	96.66	99.57	99.02	0.9944	97.94	99.25	99.02	0.9992	98.49 *	98.95 *	98.84 *	0.9968 *	93.93	99.57	98.48	0.9936
		Si	96.58	97.71	97	0.9908	98.85 *	98.41 *	98.56 *	0.9964 *	97.58	97.54	97.39	0.9971	95.01	97.87	96.36	0.9926
		All	86.97	97.09	93.60	0.9757	93.50	97.08	95.92	0.9927	95.28 *	96.52 *	96.07 *	0.9915 *	90.79	96.02	94.25	0.9731
	1.95–50 Hz	Ga	96.27	98.59	97.69	0.9939	97.77	97.10	97.25	0.9960	96.34	98.47	97.70	0.9975	98.22	97.72	97.85	0.9892
		Ju	96.09	99.14	98.57	0.9917	96.11	98.50	98.04	0.9960	94.60	98.93	98.03	0.9990	95.19	98.71	98.03	0.9860
		Si	97.04	97.03	96.87	0.9977	98.25	96.78	97.40	0.9946	94.09	97.83	95.97	0.9950	96.88	96.73	96.75	0.9893
		All	84.86	95.70	92.06	0.9645	89.21	95.84	93.69	0.9827	89.35	94.43	92.83	0.9728	83.84	95	91.28	0.9643
15 s	0.83–1.95 Hz	Ga	97.99	98.03	97.99	0.9944	99.01 *	99.17 *	99.11 *	0.9996 *	97.42	98.69	98.22	0.9971	97.37	98.54	98.11	0.9959
		Ju	94.78	98.36	97.67	0.9940	94.87	99.17	98.35	0.9975	92.38	98.87	97.53	0.9984	95.18	98.85	98.08	0.9902
		Si	95.48	97.99	96.49	0.9946	98.80	97.57	98.05	0.9961	97.05	96.60	96.69	0.9983	97.52	97.95	97.66	0.9913
		All	87.69	94.94	92.62	0.9653	90.40	96.41	94.48	0.9884	90.74	96.39	94.53	0.9875	89.53	93.87	92.52	0.9577
	1.95–50 Hz	Ga	95.49	98.84	97.66	0.9974	97.64	97.58	97.55	0.9933	95.54	97.90	96.99	0.9971	96.37	98.20	97.55	0.9936
		Ju	96.85	97.90	97.67	0.9852	93.16	96.80	96.17	0.9911	93.69	97.93	96.99	0.9848	92.06	96.63	95.62	0.9616
		Si	98.30	96.23	97.06	0.9943	94.67	96.61	95.32	0.9923	97.11	96.87	96.87	0.9950	96.27	96	95.90	0.9835
		All	89.74	92.17	91.17	0.9416	86.48	93.99	91.45	0.9660	89.73	93.14	91.73	0.9685	81.42	94.31	89.95	0.9499
30 s	0.83–1.95 Hz	Ga	93.56	98.09	96.21	0.9941	98.52	97.09	97.53	0.9979	96.28	96.17	95.98	0.9980	93.39	97.76	95.99	0.9889
		Ju	91.90	98.23	96.67	0.9883	97.50	98.58	98.20	0.9980	100	98.27	98.49	0.9980	93.81	97.18	96.37	0.9794
		Si	96.86	97.85	97.31	0.9932	94	95.14	94.52	0.9906	94.32	95.52	94.12	0.9932	95.63	94.25	94.54	0.9830
		All	91.46	94.26	93.33	0.9714	91.52	94.96	93.81	0.9892	92.38	94.94	94	0.9856	88.14	91.95	90.72	0.9310
	1.95–50 Hz	Ga	93.17	97.49	95.76	0.9928	95.08	98.68	97.32	0.9955	93.88	97.78	96.18	0.9944	91.93	97.01	95.09	0.9862
		Ju	86.21	98.54	96.09	0.9754	98.57	96.58	96.65	0.9868	95.48	96.59	96.08	0.9987	77.11	95.68	91.56	0.9370
		Si	97.63	94.40	95.32	0.9788	94.24	94.78	94.18	0.9884	93.10	97.31	94.89	0.9909	93.07	93.81	93.29	0.9767
		All	84.45	89.83	88	0.9153	86.32	93.12	90.72	0.9635	91.90	87.88	88.69	0.9421	81.50	92	87.71	0.9272

Note: * selected by Youden’s index criteria as the best classification result.

**Table 18 sensors-21-05207-t018:** Two-class classification of RF for CO and PD (Class 2, Class 2.5, and Class 3) using several CNN classifiers (AlexNet, ResNet-50, ResNet-101, and GoogLeNet) with 10-fold cross-validation.

Time Window	Frequency Range	Dataset	AlexNet				ResNet-50				ResNet-101				GoogLeNet			
			Sen (%)	Spec (%)	Acc (%)	AUC	Sen (%)	Spec (%)	Acc (%)	AUC	Sen (%)	Spec (%)	Acc (%)	AUC	Sen (%)	Spec (%)	Acc (%)	AUC
10 s	0.83–1.95 Hz	Ga	96.03	99.44	98.22	0.9983	98.47	98.48	98.44	0.9982	97.82	98.37	98.14	0.9995	97.18	98.47	98	0.9915
		Ju	95.78	99.46	98.75	0.9970	97.52	98.72	98.48	0.9966	95.66	99.35	98.66	0.9991	97.97 *	98.72 *	98.57 *	0.9895 *
		Si	96.84	98.86	97.78	0.9972	98.62 *	98.63 *	98.57 *	0.9994 *	97.84	99.07	98.44	0.9990	97.74	97.06	97.27	0.9972
		All	91.33	96.94	95.12	0.9749	92.29	97.66	95.92	0.9929	94.46 *	97.69 *	96.63 *	0.9949 *	90.05	97.06	94.78	0.9763
	1.95–50 Hz	Ga	99.56 *	98.18 *	98.59 *	0.9941 *	97.83	97.62	97.62	0.9974	93.88	98.45	96.81	0.9940	97.15	97.93	97.63	0.9922
		Ju	96.33	99.57	98.93	0.9969	93.99	98.82	97.86	0.9963	96.54	98.31	97.95	0.9993	94.58	97.87	97.23	0.9729
		Si	97.18	98.62	97.78	0.9976	98.28	96.57	97.27	0.9967	95.37	97.28	96.22	0.9958	97.49	97.05	97.13	0.9909
		All	82.20	96.42	91.41	0.9661	89.92	95.07	93.45	0.9780	89.03	94.79	92.92	0.9747	85.99	93.93	91.28	0.9590
15 s	0.83–1.95 Hz	Ga	98.03	98.56	98.32	0.9964	99	98.21	98.44	0.9976	96.90	98.04	97.55	0.9988	95.81	98.83	97.77	0.9939
		Ju	94.59	98.68	97.94	0.9946	97.18	99.03	98.63	0.9992	96.26	99.18	98.64	0.9948	95.45	97.92	97.40	0.9831
		Si	97.65	98.27	97.85	0.9950	97.93	98.26	98.04	0.9991	97.63	97.61	97.46	0.9989	98.01	98.64	98.24	0.9938
		All	90.97	96.16	94.49	0.9739	92.29	97.25	95.61	0.9914	93.91	96.42	95.60	0.9883	91.22	95.17	93.92	0.9584
	1.95–50 Hz	Ga	97.53	99.02	98.44	0.9980	98.22	96.22	96.77	0.9958	94.04	97.52	96.20	0.9942	94.67	97.58	96.43	0.9784
		Ju	94.66	98.68	97.95	0.9911	95.03	97.40	96.85	0.9919	95.29	98.40	97.53	0.9968	92.09	97.26	96.31	0.9574
		Si	96.33	97.87	97.07	0.9954	96.56	95.50	95.91	0.9914	94.20	95.30	94.33	0.9900	93.41	97.56	95.51	0.9857
		All	85.61	93.48	90.84	0.9471	88.84	95.20	93.08	0.9779	88.58	94.21	92.33	0.9716	82.67	91.90	88.78	0.9363
30 s	0.83–1.95 Hz	Ga	97.37	97.81	97.54	0.9917	94.65	98.41	96.87	0.9982	96.75	97.77	97.33	0.9982	95.13	98.06	96.85	0.9848
		Ju	97.50	98.25	97.89	0.9917	88.69	98.61	96.37	0.9936	95	97.29	96.37	0.9848	90	97.50	95.80	0.9970
		Si	93.76	95.55	94.18	0.9884	97.74	96.90	96.85	0.9930	92.77	93.67	92.94	0.9837	92.25	93.59	92.54	0.9740
		All	87.48	95.10	92.47	0.9688	90.57	96.13	94.19	0.9838	91.06	94.54	93.33	0.9862	85.24	90.09	88.49	0.9137
	1.95–50 Hz	Ga	95.60	97.47	96.64	0.9873	94.31	98.11	96.43	0.9925	93.82	98.35	96.65	0.9929	92.96	97.12	95.52	0.9828
		Ju	100	96.56	96.99	0.9767	92.86	96.86	95.81	0.9927	88.21	97.53	95.48	0.9883	88.83	96.13	94.90	0.9736
		Si	95.43	95.42	94.92	0.9913	96.25	94.19	94.54	0.9895	90.45	97.71	93.72	0.9881	94.22	94.50	94.15	0.9784
		All	79.65	93.84	88.69	0.9467	84.63	94.54	90.72	0.9684	82.25	93.65	89.65	0.9580	79.24	92.54	88	0.9250

Note: * selected by Youden’s index criteria as the best classification result.

**Table 19 sensors-21-05207-t019:** Two-class classification of CF for CO and PD (Class 2, Class 2.5, and Class 3) using several CNN classifiers (AlexNet, ResNet-50, ResNet-101, and GoogLeNet) with 10-fold cross-validation.

Time Window	Frequency Range	Dataset	AlexNet				ResNet-50				ResNet-101				GoogLeNet			
			Sen (%)	Spec (%)	Acc (%)	AUC	Sen (%)	Spec (%)	Acc (%)	AUC	Sen (%)	Spec (%)	Acc (%)	AUC	Sen (%)	Spec (%)	Acc (%)	AUC
10 s	0.83–1.95 Hz	Ga	98.03	98.79	98.52	0.9982	99.77 *	98.80 *	99.11 *	0.9995 *	99.10	98.80	98.89	0.9994	99.12	97.75	98.14	0.9875
		Ju	95.16	99.35	98.57	0.9919	96.70	99.24	98.75	0.9991	98.94 *	99.04 *	99.01 *	0.9993 *	94.31	99.24	98.31	0.9864
		Si	97	97.67	97.27	0.9969	98	97.24	97.53	0.9988	98.88 *	97.35 *	97.92 *	0.9984 *	97.81	96.35	96.87	0.9899
		All	89.22	94.71	92.98	0.9632	90.94	96.43	94.68	0.9889	92.71 *	95.11 *	94.37 *	0.9856 *	90.44	94.36	93.17	0.9533
	1.95–50 Hz	Ga	98.26	99.34	98.96	0.9976	97.26	99.02	98.36	0.9968	96.11	98.89	97.92	0.9978	97.53	97.81	97.70	0.9900
		Ju	97.19	98.94	98.57	0.9919	95.09	98.71	98.04	0.9927	95.30	98.83	98.13	0.9961	95.59	98.41	97.86	0.9758
		Si	98.65	97.49	97.92	0.9969	97.22	97.45	97.26	0.9968	95.63	99.05	97.40	0.9974	96.09	98.58	97.40	0.9934
		All	88.38	96.18	93.66	0.9757	90.85	95.88	94.19	0.9843	92.07	95.41	94.31	0.9844	90.29	96.05	94.19	0.9755
15 s	0.83–1.95 Hz	Ga	97.09	97.45	97.21	0.9899	98.37	99.35	99	0.9998	98.67	98.84	98.77	0.9991	96.90	96.92	96.88	0.9846
		Ju	95.35	98.70	98.08	0.9860	94.99	99.01	98.22	0.9990	96.43	98.69	98.22	0.9988	94.07	99.02	98.08	0.9978
		Si	93.83	97.13	95.51	0.9945	97.57	97.65	97.45	0.9950	98.23	94.94	96.28	0.9953	96.33	96.52	96.29	0.9856
		All	89.05	94.36	92.71	0.9730	88.90	96.13	93.74	0.9870	91.83	93.76	92.99	0.9842	88.15	95.42	93.03	0.9580
	1.95–50 Hz	Ga	97.27	99.50	98.67	0.9977	95.18	98.35	97.21	0.9964	96.47	98.52	97.77	0.9969	96.72	98.19	97.66	0.9941
		Ju	93.42	98.69	97.53	0.9910	95.52	96.97	96.58	0.9888	92.78	98.34	97.26	0.9967	87.30	97.07	95.21	0.9610
		Si	94.54	98.93	96.69	0.9944	97.09	96.99	96.87	0.9959	91.87	98.54	95.12	0.9951	95.76	97.19	96.31	0.9938
		All	87.20	95.01	92.47	0.9660	90.81	94.65	93.32	0.9766	90.14	94.54	93.13	0.9773	85.72	94.57	91.63	0.9650
30 s	0.83–1.95 Hz	Ga	90.96	95.61	93.51	0.9864	96.79	97.41	97.09	0.9959	97.21	95.92	96.19	0.9938	94.08	95.52	94.84	0.9863
		Ju	89.29	97.85	96.08	0.9781	96.90	96.85	96.68	0.9881	91.24	96.46	95.49	0.9853	92.74	98.24	96.98	0.9790
		Si	92.14	95.15	93.37	0.9876	97.33	98	97.23	0.9974	98.32	96.67	97.29	0.9943	94.72	96.28	94.88	0.9731
		All	85.84	93.25	90.81	0.9671	88.59	96.23	93.61	0.9865	89.86	93.22	92.17	0.9801	83.39	92.55	89.27	0.9222
	1.95–50 Hz	Ga	94.98	98.09	96.87	0.9924	96.02	96.19	95.96	0.9918	93.19	98.72	96.65	0.9924	93.31	98.05	96.20	0.9881
		Ju	96.90	98.93	98.48	0.9827	94.90	98.58	97.87	0.9981	97.14	99.30	98.80	0.9892	95.71	98.92	98.21	0.9881
		Si	96.83	93.23	94.14	0.9812	98.40	96.08	96.86	0.9923	94.14	98.08	95.72	0.9888	93.74	93.35	92.97	0.9841
		All	81.38	92.91	88.69	0.9404	86.97	94.96	92.06	0.9692	85.91	94.42	91.40	0.9670	85.33	89.91	88.30	0.9126

Note: * selected by Youden’s index criteria as the best classification result.

**Table 20 sensors-21-05207-t020:** Multi-class (CO (Class 0) vs. PD Stage 2 (Class 2) vs. PD Stage 2.5 (Class 2.5), and PD Stage 3 (Class 3)) and two-class classification (CO vs. PD (Class 2, Class 2.5, and Class 3)) summary using several CNN classifiers (AlexNet, ResNet-50, ResNet-101, and GoogLeNet) with 10-fold cross-validation for Ga dataset.

Dataset	vGRF Signal	CNN Classifier	Time Window	Frequency Range	Classification Task		Evaluation Parameters			
							Sen (%)	Spec (%)	Acc (%)	AUC
Ga	LF	GoogLeNet	15 s	0.83–1.95 Hz	Multi-Class	Class 0	97.98	99.17	98.77	0.9857
		AlexNet	10 s	1.95–50 Hz		Class 2	95.93	98.25	97.40	0.9709
		AlexNet	10 s	1.95–50 Hz		Class 2.5	93.75	98.83	97.92	0.9629
		AlexNet	30 s	1.95–50 Hz		Class 3	98.21	97.44	97.54	0.9783
		ResNet-50	15 s	0.83–1.95 Hz	Two-Class		99.01	99.17	99.11	0.9996
	RF	ResNet-101	10 s	0.83–1.95 Hz	Multi-Class	Class 0	98.43	99	98.81	0.9872
		AlexNet	10 s	1.95–50 Hz		Class 2	97.97 *	97.78 *	97.85 *	0.9787 *
		AlexNet	15 s	0.83–1.95 Hz		Class 2.5	96.88 *	96.88 *	96.88 *	0.9688 *
		ResNet-50	10 s	1.95–50 Hz		Class 3	94.05	99.24	98.59	0.9664
		AlexNet	10 s	1.95–50 Hz	Two-Class		99.56	98.18	98.59	0.9941
	CF	AlexNet	15 s	1.95–50 Hz	Multi-Class	Class 0	98.65 *	99 *	98.89 *	0.9883 *
		AlexNet	15 s	1.95–50 Hz		Class 2	96.65	98.42	97.77	0.9753
		AlexNet	15 s	1.95–50 Hz		Class 2.5	91.88	99.19	97.88	0.9553
		AlexNet	15 s	1.95–50 Hz		Class 3	99.11 *	98.98 *	99 *	0.9904 *
		ResNet50	10 s	0.83–1.95 Hz	Two-Class		99.77 *	98.80 *	99.11 *	0.9995 *

Note: * denotes the best classification result and was selected using Youden’s index criteria.

**Table 21 sensors-21-05207-t021:** Multi-class (CO (Class 0) vs. PD Stage 2 (Class 2) vs. PD Stage 2.5 (Class 2.5), and PD Stage 3 (Class 3)) and two-class classification (CO vs. PD (Class 2, Class 2.5, and Class 3)) summary using several CNN classifiers (AlexNet, ResNet-50, ResNet-101, and GoogLeNet) with 10-fold cross-validation for Ju dataset.

Dataset	vGRF Signal	CNN Classifier	Time Window	Frequency Range	Classification Task		Evaluation Parameters			
							Sen (%)	Spec (%)	Acc (%)	AUC
Ju	LF	ResNet-50	10 s	0.83–1.95 Hz	Multi-Class	Class 0	96.98	99.67	99.20	0.9833
		ResNet-50	10 s	0.83–1.95 Hz		Class 2	96.59	98.44	97.86	0.9751
		AlexNet	10 s	1.95–50 Hz		Class 2.5	97.39	96.52	96.88	0.9695
		AlexNet	30 s	0.83–1.95 Hz		Class 3	97.14	98.65	98.49	0.9790
		ResNet-101	10 s	0.83–1.95 Hz	Two-Class		98.49	98.95	98.84	0.9968
	RF	AlexNet	10 s	1.95–50 Hz	Multi-Class	Class 0	97.99 *	99.24 *	99.02 *	0.9861 *
		ResNet-101	10 s	0.83–1.95 Hz		Class 2	97.73 *	97.40 *	97.50 *	0.9756 *
		ResNet-101	10 s	0.83–1.95 Hz		Class 2.5	95.87	96.36	96.16	0.9612
		ResNet-50	15 s	0.83–1.95 Hz		Class 3	95.83	98.48	98.22	0.9716
		GoogLeNet	10 s	0.83–1.95 Hz	Two-Class		97.97	98.72	98.57	0.9895
	CF	AlexNet	10 s	1.95–50 Hz	Multi-Class	Class 0	98.49	98.91	98.84	0.9870
		ResNet-50	10 s	0.83–1.95 Hz		Class 2	96.02	98.18	97.50	0.9710
		AlexNet	10 s	1.95–50 Hz		Class 2.5	96.52 *	97.88 *	97.32 *	0.9720 *
		AlexNet	10 s	1.95–50 Hz		Class 3	100 *	99.01 *	99.11 *	0.9951 *
		ResNet-101	10 s	0.83–1.95 Hz	Two-Class		98.94 *	99.04 *	99.01 *	0.9993 *

Note: * denotes the best classification result and was selected using Youden’s index criteria.

**Table 22 sensors-21-05207-t022:** Multi-class (CO (Class 0) vs. PD Stage 2 (Class 2) vs. PD Stage 2.5 (Class 2.5), and PD Stage 3 (Class 3)) and two-class classification (CO vs. PD (Class 2, Class 2.5, and Class 3)) summary using several CNN classifiers (AlexNet, ResNet-50, ResNet-101, and GoogLeNet) with 10-fold cross-validation for Si dataset.

Dataset	vGRF Signal	CNN Classifier	Time Window	Frequency Range	Classification Task		Evaluation Parameters			
							Sen (%)	Spec (%)	Acc (%)	AUC
Si	LF	AlexNet	10 s	1.95–50 Hz	Multi-Class	Class 0	98.56 *	99.05 *	98.83 *	0.9881 *
		AlexNet	10 s	1.95–50 Hz		Class 2	95.83 *	98.84 *	97.53 *	0.9734 *
		AlexNet	10 s	1.95–50 Hz		Class 2.5	98.81 *	98.39 *	98.44 *	0.9860 *
		ResNet-50	10 s	0.83–1.95 Hz	Two-Class		98.85 *	98.41 *	98.56 *	0.9964 *
	RF	ResNet-101	10 s	0.83–1.95 Hz	Multi-Class	Class 0	99.43	97.86	98.57	0.9864
		ResNet-101	10 s	0.83–1.95 Hz		Class 2	95.83	97.22	96.61	0.9653
		ResNet-50	15 s	1.95–50 Hz		Class 2.5	94.64	98.03	97.66	0.9633
		ResNet-50	10 s	0.83–1.95 Hz	Two-Class		98.62	98.63	98.57	0.9994
	CF	AlexNet	10 s	1.95–50 Hz	Multi-Class	Class 0	97.41	99.52	98.57	0.9847
		AlexNet	10 s	1.95–50 Hz		Class 2	94.94	97.69	96.48	0.9631
		AlexNet	15 s	1.95–50 Hz		Class 2.5	98.21	97.37	97.46	0.9779
		ResNet-101	10 s	0.83–1.95 Hz	Two-Class		98.88	97.35	97.92	0.9984

Note: * denotes the best classification result and was selected using Youden’s index criteria.

**Table 23 sensors-21-05207-t023:** Multi-class (CO (Class 0) vs. PD Stage 2 (Class 2) vs. PD Stage 2.5 (Class 2.5), and PD Stage 3 (Class 3)) and two-class classification (CO vs. PD (Class 2, Class 2.5, and Class 3)) Summary using several CNN classifiers (AlexNet, ResNet-50, ResNet-101, and GoogLeNet) with 10-fold cross-validation for all datasets.

Dataset	vGRF Signal	CNN Classifier	Time Window	Frequency Range	Classification Task		Evaluation Parameters			
							Sen (%)	Spec (%)	Acc (%)	AUC
All	LF	ResNet-101	10 s	0.83–1.95 Hz	Multi-Class	Class 0	94.27 *	97.68 *	96.63 *	0.9597 *
		ResNet-101	10 s	0.83–1.95 Hz		Class 2	87.63 *	94.06 *	91.72 *	0.9085 *
		ResNet-50	10 s	0.83–1.95 Hz		Class 2.5	91.45 *	93.35 *	92.89 *	0.9240 *
		GoogLeNet	10 s	1.95–50 Hz		Class 3	93.14	98.17	97.74	0.9566
		ResNet-101	10 s	0.83–1.95 Hz	Two-Class		95.28	96.52	96.07	0.9915
	RF	ResNet-50	10 s	0.83–1.95 Hz	Multi-Class	Class 0	94.97	96.34	95.92	0.9566
		ResNet-101	10 s	0.83–1.95 Hz		Class 2	87.88	93.72	91.59	0.9080
		ResNet-50	15 s	0.83–1.95 Hz		Class 2.5	89.53	94.82	93.55	0.9218
		AlexNet	10 s	1.95–50 Hz		Class 3	94.95 *	97.30 *	97.09 *	0.9612 *
		ResNet-101	10 s	0.83–1.95 Hz	Two-Class		94.46 *	97.69 *	96.63 *	0.9949 *
	CF	ResNet-50	10 s	0.83–1.95 Hz	Multi-Class	Class 0	91.65	96.21	94.81	0.9393
		ResNet-50	10 s	0.83–1.95 Hz		Class 2	87.37	91.19	89.80	0.8928
		ResNet-50	10 s	0.83–1.95 Hz		Class 2.5	85.20	95.55	93.04	0.9038
		AlexNet	10 s	1.95–50 Hz		Class 3	92.06	98.61	98.05	0.9534
		ResNet-101	10 s	0.83–1.95 Hz	Two-Class		92.71	95.11	94.37	0.9856

Note: * denotes the best classification result and was selected using Youden’s index criteria.

**Table 24 sensors-21-05207-t024:** Multi-class classification results of comparisons with existing literature.

Literature (Year) (Cross-Validation)	Ga Dataset Acc (%)				Ju Dataset Acc (%)				Si Dataset Acc (%)		
	CO	PD Stage 2	PD Stage 2.5	PD Stage 3	CO	PD Stage 2	PD Stage 2.5	PD Stage 3	CO	PD Stage 2	PD Stage 2.5
Zhao et al. (2018) [14] (10foldCV)	100	93.33	100	100	100	100	92.31	100	100	96.55	100
Proposed Method (10foldCV)	99.03	97.85	96.87	99	98.84	97.86	97.32	99.11	98.83	97.53	98.44

**Table 25 sensors-21-05207-t025:** Two-class classification results of comparisons with existing literature for all datasets.

Literature (Year)	Cross-Validation	Evaluation Parameters			
		Sen (%)	Spec (%)	Acc (%)	AUC
Maachi et al. [39] (2020)	10foldCV	98.10	100	98.70	-
Wu et al. [40] (2017)	LOOCV	72.41	96.55	84.48	0.9049
Ertugrul et al. [41] (2016)	10foldCV	88.90	82.20	88.89	-
Zeng et al. [42] (2016)	5foldCV	96.77	95.89	96.39	-
Daliri [43] (2013)	50% training 50% testing	91.71	89.92	91.20	-
Proposed Method	10foldCV	94.46	97.69	96.63	0.9949

**Table 26 sensors-21-05207-t026:** Two-class classification results of comparisons with existing literature for each sub-dataset.

Literature (Year)	Cross-Validation	Ga Dataset Acc (%)	Ju Dataset Acc (%)	Si Dataset Acc (%)
Khoury et al. [44] (2019)	LOOCV	86.05	90.91	82.81
Khoury et al. [45] (2018)	10foldCV	93.57	97.52	87.22
Proposed Method	10foldCV	99.11	99.01	98.56

## Data Availability

Not applicable.

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
