# Peer review of "Implementation of a Deep Learning Algorithm Based on Vertical Ground Reaction Force Time–Frequency Features for the Detection and Severity Classification of Parkinson’s Disease"

_sensors, 2021, doi:10.3390/s21155207_

Round 1
Reviewer 1 Report
The authors presented methods to detect and classify Parkinson's disease using deep learning algorithms. The topic is interesting and meaningful to the society. Overall the manuscript is well organized and written. Evaluation has been done to compare the performance with different networks. Experiment design was proper and results seem to be promising. A few suggestions which may further improve the manuscript.
- The sub-section numbering seems incorrect under 2.5.
- Can the authors add more discussions regarding the advantage / disadvantages of the networks, including the possible reasons for the difference in performance?
Reviewer 2 Report
This paper presents a model for parkinson disease detection. I think paper is interesting after revisions:
- What are limitation of your system? Detection precision and accuracy of sensor readings are crucial.
- How did you select phases in fig. 1? Did you consult them with medicians?
- How did you select gaits for this type movement?
- What should be optimal frequency of readings during movement for your system?
- System needs comparisons to other solutions.
Reviewer 3 Report
The article presents an interesting approach to the detection and severity classification of Parkinson’s disease. It is clearly written with only minor typos that can be easily corrected in final editing.
The introduction section presents an adequate review of other works in the area and the materials and methods section does a good job of introducing the reader to the different techniques used in this work. The proposed approach is clearly described, with enough information to allow replication.
The experimental setup is appropriate, with plentiful experimental results. The overall results are promising and seem to support this method’s ability to effectively detect and classify the severity of Parkinson’s disease. The conclusions presented by the authors are also well supported by the results and their discussion.
Overall, I believe the article presents an original approach, clearly explained and supported by the experimental results and can be published with only minor corrections to the text.
